# TUVF: Learning Generalizable Texture UV Radiance Fields

**An-Chieh Cheng**[1]  **Xueting Li**[2]  **Sifei Liu**[2†]  **Xiaolong Wang**[1†]
[1]UC San Diego  [2]NVIDIA

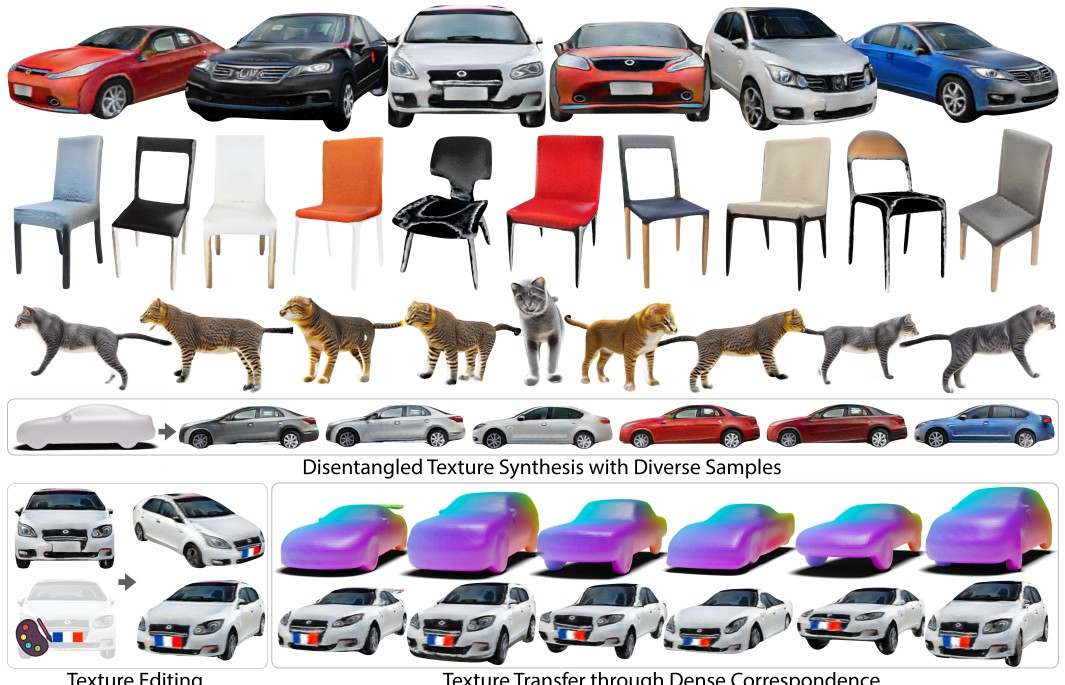

Disentangled Texture Synthesis with Diverse Samples

Texture Editing     Texture Transfer through Dense Correspondence

Figure 1: We propose Texture UV Radiance Fields (TUVF) to render a 3D consistent texture given a 3D object shape input. TUVF provides a category-level texture representation disentangled from 3D shapes. **Top three rows:** TUVF can synthesize realistic textures by training from a collection of single-view images; **Fourth row:** Given a 3D shape input, we can render different textures on top by using different texture codes; **Bottom row:** We can perform editing on a given texture (adding a flag of France) and directly apply the same texture on different 3D shapes without further fine-tuning. Note that all samples are rendered under 1024×1024 resolution; zoom-in is recommended.

## ABSTRACT

Textures are a vital aspect of creating visually appealing and realistic 3D models. In this paper, we study the problem of generating high-fidelity texture given shapes of 3D assets, which has been relatively less explored compared with generic 3D shape modeling. Our goal is to facilitate a controllable texture generation process, such that one texture code can correspond to a particular appearance style independent of any input shapes from a category. We introduce Texture UV Radiance Fields (TUVF) that generate textures in a learnable UV sphere space rather than directly on the 3D shape. This allows the texture to be disentangled from the underlying shape and transferable to other shapes that share the same UV space, i.e., from the same category. We integrate the UV sphere space with the radiance field, which provides a more efficient and accurate representation of textures than traditional texture maps. We perform our experiments on synthetic and real-world object datasets where we achieve not only realistic synthesis but also substantial improvements over state-of-the-arts on texture controlling and editing. We show more qualitative results at https://www.anjiecheng.me/TUVF.

† Equal advising.

Codes and trained models are publicly available at https://github.com/AnjieCheng/TUVF.

# 1 INTRODUCTION

3D content creation has attracted much attention given its wide applications in mixed reality, digital twins, filming, and robotics. However, while most efforts in computer vision and graphics focus on 3D shape modeling (Chabra et al., 2020; Zeng et al., 2022; Cheng et al., 2023), there is less emphasis on generating realistic textures (Siddiqui et al., 2022). Textures play a crucial role in enhancing the immersive experiences in virtual and augmented reality. While 3D shape models are abundant in simulators, animations, video games, industry manufacturing, synthetic architectures, etc., rendering realistic and 3D consistent texture on these shape models without human efforts (Figure 1 first three rows) will fundamentally advance the visual quality, functionalities, and experiences.

Given instances of one category, ideally, their textures should be disentangled from their shapes. This can be particularly useful in scenarios where the appearance of an object needs to be altered frequently, but the shape remains the same. For example, it is common in video games to have multiple variations of the same object with different textures to provide visual variety without creating entirely new 3D models. Thus, the synthesis process should also be controllable, i.e., we can apply different textures to the exact shape (Figure 1 fourth row) or use the same texture code for different shapes and even edit part of the texture (Figure 1 bottom row). Recently, the wide utilization of GANs (Goodfellow et al., 2014) allows training on 3D content creation with only 2D supervision (Nguyen-Phuoc et al., 2019; Siddiqui et al., 2022; Skorokhodov et al., 2022; Chan et al., 2022). While this alleviates the data and supervision problem, the learned texture representation often highly depends on the input geometry, making the synthesis process less controllable: With the same texture code or specifications, the appearance style of the generated contents changes based on the geometric inputs.

We propose a novel texture representation, Texture UV Radiance Fields (TUVF), for high-quality and disentangled texture generation on a given 3D shape, i.e., a sampled texture code represents a particular appearance style adaptable to different shapes. The key to disentangling the texture from geometry is to generate the texture in a canonical UV sphere space instead of directly on the shape. We train the canonical UV space for each category via a Canonical Surface Auto-encoder in a self-supervised manner so that the correspondence between the UV space and the 3D shape is automatically established during training. Unlike traditional UV mesh representation, TUVF does not suffer from topology constraints and can easily adapt to a continuous radiance field.

Given a texture code, we first encode it with a texture mapping network to a style embedding, which is then projected onto the canonical UV sphere as a textured UV sphere. Using correspondence, we can assign textures to arbitrary 3D shapes and construct a point-based radiance field. Consequently, we sample the points along the ray and around the object shape surface and render the RGB image. In contrast to volumetric rendering (Drebin et al., 1988; Mildenhall et al., 2020), our Texture UV Radiance Field allows efficient rendering and disentangles the texture from the 3D surface. Finally, we apply an adversarial loss using high-quality images from the same category.

We train our model on two real-world datasets (Yang et al., 2015; Park et al., 2018a), along with a synthetic dataset generated by our dataset pipeline. Figure 1 visualizes the results of synthesizing 3D consistent texture given a 3D shape. Our method can provide realistic texture synthesis. More importantly, our method allows complete texture disentanglement from geometry, enabling controllable synthesis and editing (Figure 1 bottom two rows). With the same shape, we evaluate how diverse the textures can be synthesized. With the same texture, we evaluate how consistently it can be applied across shapes. Our method outperforms previous state-of-the-arts significantly on both metrics.

# 2 RELATED WORK

**Neural Radiance Fields.** Neural Radiance Fields (NeRFs) have been widely studied on broad applications such as high fidelity novel view synthesis (Mildenhall et al., 2020; Barron et al., 2021; 2023) and 3D reconstruction (Wang et al., 2021; Yariv et al., 2021; Zhang et al., 2021). Following this line of research, the generalizable versions of NeRF are proposed for faster optimization and few-view synthesis (Schwarz et al., 2020; Trevithick & Yang, 2021; Li et al., 2021a; Chen et al., 2021; Wang et al., 2022; Venkat et al., 2023). Similarly, TUVF is trained in category-level and learn across instances. However, instead of learning from reconstruction with multi-view datasets (Yu et al., 2021a; Chen et al., 2021), our method leverages GANs for learning from 2D single-view image collections. From the rendering perspective, instead of performing volumetric rendering (Drebin et al., 1988), more efficient rendering techniques have been applied recently, including surface rendering (Niemeyer et al., 2020; Yariv et al., 2020) and rendering with point clouds (Xu et al., 2022;

Yang et al., 2022; Zhang et al., 2023b). Our work relates to the point-based paradigm: Point-Nerf (Xu et al., 2022) models a volumetric radiance field using a neural point cloud; Neu-Mesh (Yang et al., 2022) proposes a point-based radiance field using mesh vertices. However, these approaches typically require densely sampled points and are optimized for each scene. In contrast, TUVF only requires sparse points for rendering and is generalizable across scenes.

**Texture Synthesis on 3D Shapes.** Texture synthesis has been an active research area in computer vision and graphics for a long time, with early works focusing on 2D image textures (Cross & Jain, 1983; Taubin, 1995; Efros & Leung, 1999) and subsequently expanding to 3D texture synthesis (Turk, 2001; Bhat et al., 2004; Kopf et al., 2007). Recently, learning-based methods (Raj et al., 2019; Siddiqui et al., 2022; Foti et al., 2022) combined with differentiable rendering techniques (Liu et al., 2019; Mildenhall et al., 2020) have shown promising results in texture synthesis on 3D shapes by leveraging generative adversarial networks (GANs) (Goodfellow et al., 2014) and variational autoencoders (VAEs) (Kingma & Welling, 2013). These paradigms have been applied to textured shape synthesis (Pavllo et al., 2020; Gao et al., 2022; Chan et al., 2022) and scene completion (Dai et al., 2021; Azinović et al., 2022). Motivated by these works, we also adopt GANs to supervise a novel representation for 3D texture synthesis. This allows our model to train from a collection of single-view images instead of using multi-view images for training.

**Texture Representations.** Several mesh-based methods (Oechsle et al., 2019; Dai et al., 2021; Yu et al., 2021b; Chen et al., 2022; Siddiqui et al., 2022; Chen et al., 2023; Yu et al., 2023) have been proposed. AUV-Net (Chen et al., 2022) embed 3D surfaces into a 2D aligned UV space using traditional UV mesh; however, they requires shape-image pairs as supervision. Texturify (Siddiqui et al., 2022) use 4-RoSy fields (Palacios & Zhang, 2007) to generate textures on a given mesh. However, the texture representation is entangled with the input shape, and the style can change when given different shape inputs. Our approach falls into the NeRF-based methods (Chan et al., 2022; Skorokhodov et al., 2022). The tri-plane representation has been widely used in these methods. However, these methods often face a similar problem in structure and style entanglement. NeuTex (Xiang et al., 2021) provides an explicit disentangled representation. However, the representation is designed for a single scene. Our TUVF representation disentangles texture from geometry and is generalizable across instances, which allows transferring the same texture from one shape to another.

**Disentanglement of Structure and Style.** The disentanglement of structure and style in generative models allows better control and manipulation in the synthesis process. Common approaches to achieve disentanglement include using Autoencoders (Kingma & Welling, 2013; Kulkarni et al., 2015; Jha et al., 2018; Mathieu et al., 2019; Liu et al., 2020; Park et al., 2020; Pidhorskyi et al., 2020) and GANs (Chen et al., 2016; Huang et al., 2018; Karras et al., 2019; Singh et al., 2019; Nguyen-Phuoc et al., 2019; Chan et al., 2021). For example, the Swapping Autoencoder (Park et al., 2020) learns disentanglement by leveraging network architecture bias and enforcing the texture branch of the network to encode co-occurrent patch statistics across different parts of the image. However, these inductive biases do not ensure full disentanglement, and the definition of disentanglement itself is not clearly defined. In the second paradigm with adversarial learning, StyleGAN (Karras et al., 2019) learns separate mappings for the structure and style of images, allowing for high-quality image synthesis with fine-grained control over image attributes. Recently, CoordGAN (Mu et al., 2022) shows that it is possible to train GANs and pixel-wise dense correspondence can automatically emerge. Our work leverages GANs to provide supervision in training, but instead of disentangling texture from 2D structures, we are learning the texture for 3D object shapes.

## 3  TEXTURE UV RADIANCE FIELDS

We introduce **T**exture **UV** Radiance **F**ields (TUVF) that generate a plausible texture UV representation conditioned on the shape of a given 3D object. Semantically corresponding points on different instances across the category are mapped to the same locations on the texture UV, which inherently enables applications such as texture transfer during inference. As shown in Figure 2, our texture synthesis pipeline begins with a canonical surface auto-encoder (Section 3.1) that builds dense correspondence between a canonical UV sphere and all instances in a category. Such dense correspondence allows us to synthesize textures on a shared canonical UV space using a coordinate-based generator (Section 3.2). Finally, since we do not assume known object poses for each instance, we render the generated radiance field (Section 3.3) and train the framework with adversarial learning (Section 3.4).

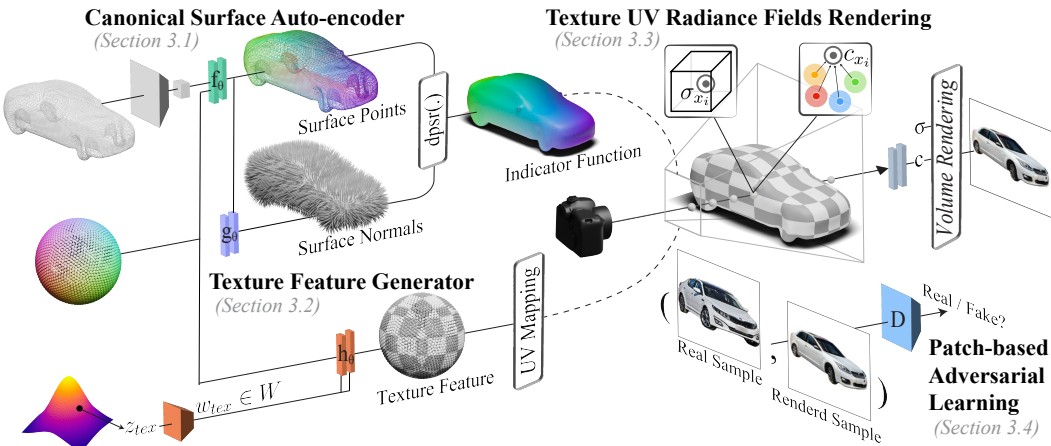

Figure 2: **Method overview**. We perform two-stage training: (i) We first train the Canonical Surface Auto-encoder (Equation 6), which learns decoders $f_\theta$ (🟩) and $g_\theta$ (🟪) predicting the coordinates and normals for each point on the UV sphere, given an encoded shape. (ii) We then train the Texture Feature Generator $h_\theta$ (🟧) which outputs a textured UV sphere. We can construct a Texture UV Radiance Field with the outputs from $f_\theta$, $g_\theta$, and $h_\theta$, and render an RGB image as the output. We perform patch-based generative adversarial learning (Equation 7) to supervise $h_\theta$.

## 3.1 CANONICAL SURFACE AUTO-ENCODER

The key intuition of this work is to generate texture on a shape-independent space, where we resort to a learnable UV space containing dense correspondences across different instances in a category. To this end, we learn a canonical surface auto-encoder that maps any point on a canonical UV sphere to a point on an object's surface (Cheng et al., 2021; 2022). Specifically, given a 3D object with point $O$, we first encode its shape into a geometry code $z_{geo} \in R^d$ by an encoder $\mathcal{E}$ (Cheng et al., 2021). For a point $p$ on the canonical UV sphere, we feed the concatenation of its coordinates $X_p$ and the geometry code into an implicit function $f_\theta$ (Figure 2 🟩) to predict the coordinates of the mapped point $p'$, denoted as $X_{p'}$, on the given object's surface. We further predict the normal $N_{p'}$ at $p'$ with a separate implicit function $g_\theta$ (Figure 2 🟪). The overall process can be denoted as follows:

$$z_{geo} = \mathcal{E}(O) \tag{1}$$

$$X_{p'} = f_\theta(X_p; z_{geo}), \quad N_{p'} = g_\theta(X_{p'}; z_{geo}) \tag{2}$$

The coordinates and normal of $p'$ are then used for the rendering process discussed in Section 3.3.

We use a graph-based point encoder following DGCNN (Wang et al., 2019) and decoder architecture following (Cheng et al., 2022) for $f_\theta$ and $g_\theta$. As proved by (Cheng et al., 2021), correspondences emerge naturally during training, and $f_\theta$ and $g_\theta$ are trained end-to-end using Chamfer Distance (Borgefors, 1988) on the surface points and the L2 losses on the indicator grid discussed in Section 3.4.

## 3.2 TEXTURE FEATURE GENERATOR

The canonical UV sphere defines dense correspondences associated with all instances in a category. Thus, shape-independent textures can be formulated as generating texture features on top of this sphere space. To this end, we introduce CIPS-UV, an implicit architecture for texture mapping function $h_\theta$ (Figure 2 🟧). Specifically, CIPS-UV takes a 3D point $p$ on the canonical sphere $X_p$, and a randomly sampled texture style vector $z_{tex} \sim \mathcal{N}(0, 1)$ as inputs and generates the texture feature vector $c_j \in R^d$ at point $p$, which are further used for rendering as discussed in Section 3.3. The style latent is injected via weight modulation, similar to StyleGAN (Karras et al., 2019). We design our $h_\theta$ based on the CIPS generator (Anokhin et al., 2021), where the style vector $z_{tex}$ is used to modulate features at each layer. This design brings two desired properties. First, combined with the canonical UV sphere, we do not require explicit parameterization, such as unwrapping to 2D. Second, it does not include operators (e.g., spatial convolutions (Schmidhuber, 2015), up/downsampling, or self-attentions (Zhang et al., 2019)) that bring interactions between pixels. This is important because nearby UV coordinates may not correspond to exact neighboring surface points in the 3D space. As a result, our generator can better preserve the 3D semantic information and produce realistic and diverse textures on the UV sphere. Please refer to Appendix L for implementation details.

### 3.3 RENDERING FROM UV SPHERE

**Efficient Ray Sampling.** Surface rendering is known for its speed, while volume rendering is known for its better visual quality (Oechsle et al., 2021). Similar to (Oechsle et al., 2021; Yariv et al., 2021; Wang et al., 2021), we take advantage of both to speed up rendering while preserving the visual quality, i.e., we only render the color of a ray on points near the object's surface. To identify valid points near the object's surface, we start by uniformly sampling 256 points along a ray between the near and far planes and computing the density value $\sigma_i$ (discussed below) for each position $x_i$. We then compute the contribution (denoted as $w_i$) of $x_i$ to the ray radiance as

$$w_i = \alpha_i \cdot Ti, \quad \alpha_i = 1 - exp(-\sigma_i \delta), \quad T_i = exp(-\sum_{j=1}^{i-1} \alpha_j \delta_j) \tag{3}$$

where $\delta$ is the distance between adjacent samples. If $w_i = 0$, then $x_i$ is an invalid sample (Hu et al., 2022) and will not contribute to the final ray radiance computation. Empirically, we found that sampling only three points for volume rendering is sufficient. It is worth noting that sampling $\sigma_i$ alone is also fast since the geometry is known in our setting.

**Volume Density from Point Clouds.** We discuss how to derive a continuous volume density from the Canonical Surface Auto-encoder (Section 3.1), which was designed to manipulate discrete points. Given a set of spatial coordinates and their corresponding normal derived from $f_\theta$ and $g_\theta$, we use the Poisson Surface Reconstruction algorithm (Peng et al., 2021) to obtain indicator function values over the 3D grid. We then retrieve the corresponding indicator value $dpsr(x_i)$ for each location $x_i$ via trilinear interpolation. $dpsr(x_i)$ is numerically similar to the signed distance to the surface and can serve as a proxy for density in volume rendering. We adopt the formulation from VolSDF (Yariv et al., 2021) to transform the indicator value into density fields $\sigma$ by:

$$\sigma(x_i) = \frac{1}{\gamma} \cdot Sigmoid(\frac{-dpsr(x_i)}{\gamma}) \tag{4}$$

Note that the parameter $\gamma$ controls the tightness of the density around the surface boundary and is a learnable parameter in VolSDF. However, since our geometry remains fixed during training, we used a fixed value of $\gamma = 5e^{-4}$.

**Point-based Radiance Field.** To compute the radiance for a shading point $x_i$, we query the $K$ nearest surface points $p'_{j \in N_K}$ in the output space of the Canonical Surface Auto-encoder and obtain their corresponding feature vector $c_{j \in N_K}$ by $h_\theta$. We then use an MLP$_F$, following (Xu et al., 2022), to process a pair-wise feature between the shading point $x_i$ and each nearby neighboring point, expressed as $c_{j,x_i} = \text{MLP}_F(c_j, p'_j - x_i)$. Next, we apply inverse distance weighting to normalize and fuse these $K$ features into one feature vector $c_{x_i}$ for shading point $x_i$:

$$c_{x_i} = \sum_{j \in N_K} \frac{\rho_j}{\sum \rho_j} c_{j,x_i}, \quad \text{where } \rho_j = \frac{1}{\|p'_j - x_i\|} \tag{5}$$

Then we use another MLP$_C$ to output a final color value for point $x_i$ based on $c_{j,x_i}$ and an optional viewing direction $d$, denoted $c_i = MLP_C(c_{j,x_i} \bigoplus d)$. We design MLP$_F$ and MLP$_C$ to be shared across all points, i.e., as implicit functions, so that they do not encode local geometry information. Finally, following quadrature rules (Mildenhall et al., 2020; Max, 1995), we render the pixel $C(\boldsymbol{r})$ with points $\{\mathbf{x}_i | i = 1, ..., N\}$ along the ray $\boldsymbol{r}$ as $\hat{C}(\boldsymbol{r}) = \sum_{i=1}^{N} T_i \alpha_i \mathbf{c}_i$.

### 3.4 GENERATIVE ADVERSARIAL LEARNING

**Patch-based Discriminator.** NeRF rendering is expressive but can be computationally expensive when synthesizing high-resolution images. For GAN-based generative NeRFs, using 2D convolutional discriminators that require entire images as inputs further exacerbates this challenge. Thus, in our work, we adopt the efficient and stable patch-discriminator proposed in EpiGRAF (Skorokhodov et al., 2022). During training, we sample patches starting from a minimal scale, covering the entire image in low resolution. As the scale gradually grows, the patch becomes high-resolution image crops. As our rendering process is relatively lightweight (see Section 3.3), we use larger patches (128×128) than those used in EpiGRAF (64×64), which brings better quality.

**Training Objectives.** We train the Canonical Surface Auto-encoder (Section 3.1) and the Texture Generator (Section 3.2) in separate stages. In stage-1, we adopt a Chamfer Distance between the output and input point sets, and a L2 loss to learn the mapping $dpsr(.)$ between points and volume density, as aforementioned:

$$L_{CSAE} = L_{CD}(p, p'_i) + L_{DPSR} \| \chi' - \chi \| \tag{6}$$

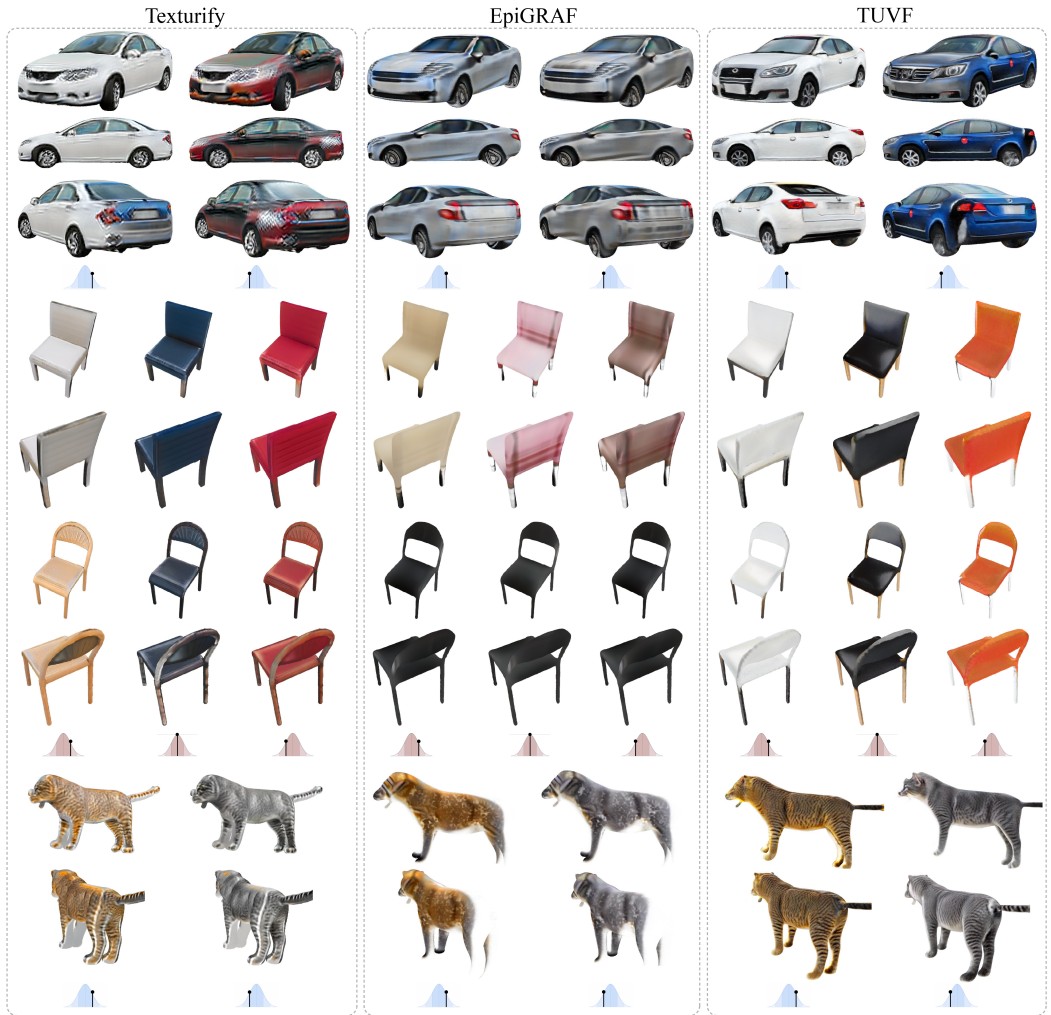

Figure 3: **Qualitative comparison.** TUVF achieves much more realistic, high-fidelity, and diverse 3D consistent textures compared to previous approaches. Each column also presents results generated using the same texture code. Texturify and EpiGRAF both have *entangled* texture and geometry representations, which occasionally result in identical global colors or similar local details despite having different texture codes. Visualized under 1024×1024 resolution; zoom-in is recommended.

where $(\chi', \chi)$ denotes the predicted and ground truth indicator function (see details in (Peng et al., 2021)). In stage-2, with $\mathbb{R}$ denotes rendering, we enforce a non-saturating GAN loss with R1 regularization to train the texture generator (Karras et al., 2019; Skorokhodov et al., 2022):

$$L_{GAN} = \mathrm{E}_{z_{tex} \sim p_G}[f(D(\mathbb{R}(h_\theta(z_{tex}, X_p))))] + \mathrm{E}_{I_{real} \sim p_D}[f(-D(I_{real}) + \lambda \|\Delta D(I_{real})\|^2], \quad (7)$$
$$where \quad f(u) = -log(1 + \exp{(-u)}).$$

## 4 EXPERIMENTS

### 4.1 DATASETS

**CompCars & Photoshape.** We used 3D shapes from ShapeNet's "chair" and "car" categories (Chang et al., 2015). For the 2D datasets, we employed Compcars (Yang et al., 2015) for cars and Photoshape (Park et al., 2018b) for chairs. Notably, the chair category includes subsets with significantly different topologies, such as lounges and bean chairs, where finding a clear correspondence may be challenging even for humans. Consequently, we evaluate our model and the baselines' performance on the "straight chair" category, one of the largest subsets in the chair dataset. For fair comparisons, we follow Texturify, splitting the 1,256 car shapes into 956 for training and 300 for testing. We apply the same split within the subset for the chair experiment, yielding 450 training and 150 testing shapes. We also provide the evaluation of full Photoshape in Appendix F.

Table 1: **Quantitative Results on CompCars.** The symbol "†" denotes an instance-specific approach, whereas the remaining methods employ category-wise training. Our method significantly improves over all previous methods on all metrics. KID is multiplied by $10^2$.

| Method | LPIPS$_g$ ↑ | LPIPS$_t$ ↓ | FID ↓ | KID ↓ |
|---|---|---|---|---|
| TexFields (Oechsle et al., 2019) | - | - | 177.15 | 17.14 |
| LTG (Yu et al., 2021b) | - | - | 70.06 | 5.72 |
| EG3D-Mesh (Chan et al., 2022) | - | - | 83.11 | 5.95 |
| Text2Tex (Chen et al., 2023)† | - | - | 46.91 | 4.35 |
| Texturify (Siddiqui et al., 2022) | 9.75 | 2.46 | 59.55 | 4.97 |
| EpiGRAF (Skorokhodov et al., 2022) | 4.26 | 2.34 | 89.64 | 6.73 |
| TUVF | 15.87 | 1.95 | 41.79 | 2.95 |

**DiffusionCats.** The above real-world dataset assumes known camera pose distributions, such as hemispheres. However, aligning in-the-wild objects into these specific poses can be time-consuming and prone to inaccuracies. Therefore, we introduce a data generation pipeline that directly synthesizes realistic texture images. We render depth maps from 3D shapes and convert these depth maps into images using pre-trained diffusion models (Rombach et al., 2022; Zhang et al., 2023a). Next, we determine the bounding box based on the depth map and feed this into an off-the-shelf segmentation model (Kirillov et al., 2023) to isolate the target object in the foreground. This pipeline eliminates the need for TUVF to depend on real-world image datasets, making it adaptable to other categories with controllable prompts. For quantitative evaluation, we use 250 shapes of cats from SMAL (Zuffi et al., 2017), which includes appearance variations and deformations, to create a new 2D-3D dataset for texture synthesis through our data generation pipeline. We discuss the details of this pipeline in Appendix P and samples of the generated dataset in Appendix Q.

## 4.2 BASELINES

**Mesh-based Approaches.** Texturfiy (Siddiqui et al., 2022) is a state-of-the-art prior work on texture synthesis. They proposed using a 4-rosy field as a better representation for meshes. TexFields (Oechsle et al., 2019), SPSG (Dai et al., 2021), LTG (Yu et al., 2021b), and EG3D-Mesh (Chan et al., 2022) are all mesh-based baselines. These baselines follow a similar framework, where the texture generators are conditioned on a certain shape geometry condition. The biggest difference among these methods is that they use different representations. Specifically, the TexFields (Oechsle et al., 2019) uses a global implicit function to predict texture for mesh, and SPSG (Dai et al., 2021) uses 3D convolution networks to predict voxels for textures. EG3D-Mesh (Chan et al., 2022) uses the triplane representation in EG3D (Chan et al., 2022) to predict the face colors for a given mesh. Note that all baselines require explicit geometry encoding for texture synthesis. On the other hand, our method relies on correspondence and does not directly condition texture on a given geometry. Furthermore, our learned dense surface correspondence allows for direct texture transfer. We also compare with a concurrent work, Text2Tex (Chen et al., 2023), which proposes an instance-specific approach for texture synthesis using a pre-trained diffusion model.

**NeRF-based Approach.** We also evaluate our method against a state-of-the-art NeRF-based approach, EpiGRAF (Skorokhodov et al., 2022), which employs a tri-plane representation and a patch-based discriminator. To modify EpiGRAF into a texture generator for radiance fields, we follow TexFields (Oechsle et al., 2019) and use a point cloud encoder to encode geometry information into EpiGRAF's style-based triplane generator. For fair comparison, we employ the same discriminator and training hyper-parameters for EpiGRAF (Skorokhodov et al., 2022) and our method.

## 4.3 EVALUATION METRICS

**LPIPS$_g$ and LPIPS$_t$.** We introduce two metrics to evaluate our model's ability to disentangle geometry and texture. The first metric, LPIPS$_g$, is calculated by generating ten random latent codes for each shape in the test set and measuring the diversity of the synthesized samples. If the model struggles to disentangle, the generated samples may appear similar, leading to a lower LPIPS score. For the second metric, LPIPS$_t$, we measure the semantic consistency after texture swapping. Specifically, we randomly sample four latent codes and transfer them among 100 test shapes. If a model successfully disentangled the geometry and texture, all samples with the same texture code should look semantically similar, leading to a lower LPIPS score.

**FID and KID.** In addition to LPIPS$_g$ and LPIPS$_t$, we employ two standard GAN image quality and diversity metrics, specifically the Frechet Inception Distance (FID) and Kernel Inception Distance

Table 2: **Quantitative Results on Photoshape and DiffusionCats.** While our method has slightly larger FID and KID than Texturify on Photoshape, we achieve significantly better results in controllable synthesis. On the other hand, our method achieves better results in both visual quality and controllable synthesis on DiffusionCats. KID is multiplied by $10^2$.

| | Photoshape | | | | DiffusionCats | | | |
|---|---|---|---|---|---|---|---|---|
| | LPIPS$_g$ ↑ | LPIPS$_t$ ↓ | FID ↓ | KID ↓ | LPIPS$_g$ ↑ | LPIPS$_t$ ↓ | FID ↓ | KID ↓ |
| EpiGRAF (Skorokhodov et al., 2022) | 7.00 | 3.14 | 65.62 | 4.20 | 5.37 | 1.98 | 196.01 | 19.10 |
| Texturify (Siddiqui et al., 2022) | 6.74 | 2.89 | 45.92 | 2.61 | 4.77 | 2.61 | 72.43 | 5.61 |
| TUVF | 14.93 | 2.55 | 51.29 | 2.98 | 11.99 | 1.50 | 64.13 | 3.56 |

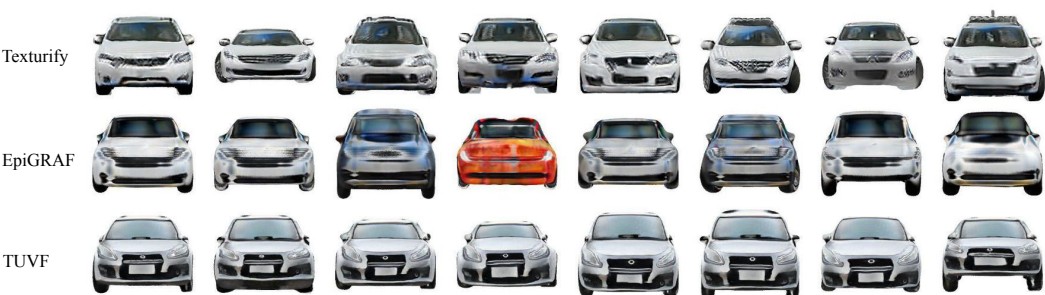

Figure 4: **Texture Transfer Comparison.** Each approach applies the same texture code to synthesize textures on different shapes. TUVF can obtain consistent textures across all shapes, while previous approaches output different styles on different object shapes when using the same texture code.

(KID) scores. We follow Texturify's setup in all experiments, training on $512 \times 512$ resolution images and rendering images at a resolution of $512 \times 512$ and subsequently downsampling to $256 \times 256$ for evaluation. We employ four random views and four random texture codes for all evaluations and incorporate all available images in the FID/KID calculations.

### 4.4 RESULTS

**Canonical Surface Auto-encoder.** To the best of our knowledge, our work represents the first attempt to explore joint end-to-end canonical point auto-encoder (Cheng et al., 2021) and surface learning (Peng et al., 2021). A key concern is the smoothness of the learned correspondence and the reconstructed surface. We construct and visualize the mesh using the predicted indicator function $\chi'$, and without requiring any proxy function (such as nearest neighbor search), dense surface correspondence is readily obtained. As a result of the Poisson surface reconstruction, $P'_j$, which holds the correspondence, naturally lies on the surface. In Figure 5 and Figure 7, we showcase that our reconstructed surface is indeed smooth and that the correspondence is both dense and smooth as well.

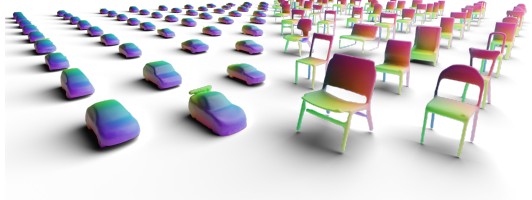

Figure 5: **Surface reconstruction with dense correspondence.** The color map indicates the correspondence between each instance and the UV. Please refer to Appendix A for further results.

**Quantitative Texture Synthesis Results.** We show the quantitative results on CompCars in Table 1, the results on Photoshape and DiffusionCats in Table 2. For the CompCars and DiffusionCats datasets, we achieve significant improvements over all the metrics. For the Photoshape dataset, while our approach is slightly worse than Texturify in FID and KID, as for the fidelity metrics, we obtain much better results on controllable synthesis. We further conduct a user study to evaluate the texture quality. Two metrics are considered: (1) General: The users compare random renders from baselines and our method, choosing the most realistic and high-fidelity method. (2) Transfer: The users compare three random renders with the same texture code, selecting the most consistent across shapes. We use Amazon Turk to collect 125 user feedback; the results are shown in Table 3.

**Qualitative Texture Synthesis Results.** We show our qualitative results for texture synthesis in Figure 3, which confirms that textures generated by our approach are more visually appealing, realistic, and diverse. EpiGRAF suffers from the redundancy of tri-plane representation, leading to less sharp results. We also observe that the tri-plane representation fails when objects are thin (e.g., cats). Our proposed method also shows better diversity and disentanglement than Texturify

Table 3: **User study.** Percentage of users who favored our method over the baselines in a user study with 125 responses.

| Dataset | Metric | Texturify ↑ | EpiGRAF ↑ |
|---------|--------|-------------|-----------|
| CompCars | General | 82.40 | 85.60 |
| | Transfer | 75.20 | 78.40 |
| Photoshape | General | 74.40 | 80.00 |
| | Transfer | 70.40 | 75.20 |

Table 4: **Ablation with different architecture designs for texture mapping function** Evaluated on CompCars. KID is multiplied by $10^2$.

| Architecture | FID ↓ | KID ↓ |
|--------------|-------|-------|
| CIPS-2D (Anokhin et al., 2021) | 148.09 | 13.38 |
| StyleGAN2 (Karras et al., 2020b) | 103.62 | 7.89 |
| CIPS-UV *(ours)* | 41.79 | 2.95 |

and EpiGRAF. We show texture transfer results in Figure 4, where Texturify and EpiGRAF failed to transfer the texture on some samples. Please refer to Appendix B for more texture synthesis results.

**Ablation study.** We conducted an ablation study on different texture generator architectures using the CompCars dataset. Two architectures were considered: CIPS-2D and StyleGAN2, the former being the same as the one proposed in (Anokhin et al., 2021), and the latter being a popular choice for both 2D and 3D GANs. Since the input to both generators is in 3D (i.e., sphere coordinates), an equirectangular projection was first performed to transform the coordinates into 2D. We show the results in Table 4, where CIPS-2D suffers from the explicit parameterization of unwrapping 3D to 2D. Similarly, StyleGAN2 suffers from pixel-wise interaction operators that degrade its performance as well. In contrast, our proposed generator design avoids explicit parameterization and operators that bring interactions between pixels. By preserving 3D semantic information, our generator produces realistic and diverse textures on the UV sphere. Please refer to Appendix H for more ablation studies.

**Texture Editing.** Our method enables texture editing (Yang et al., 2022; Bao et al., 2023) by allowing direct modification of rendered images, such as drawing or painting. Given a synthesized texture, one can directly operate on the rendered view to edit the texture. By fine-tuning the edited image through back-propagation to the texture feature, we can obtain an edited texture that is 3D consistent across different views. As shown in Figure 6, after editing an image, we can fine-tune its texture feature and transfer it to different shapes.

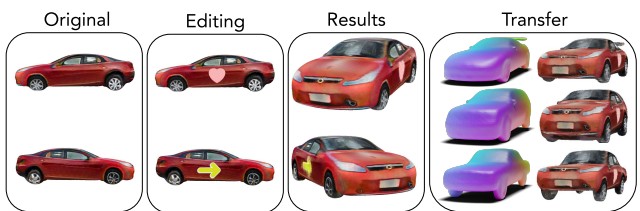

Figure 6: **Editing and Transfer Results.** The disentanglement ensures that the radiance field is independent of density conditions, enabling us to fine-tune UV texture features with sparse views. Please refer to Appendix C for more samples.

## 5 CONCLUSION, LIMITATIONS, AND FUTURE WORK

In this paper, we introduce Texture UV Radiance Fields for generating versatile, high-quality textures applicable to a given object shape. The key idea is to generate textures in a learnable UV sphere space independent of shape geometry and compact and efficient as a surface representation. Specifically, we leverage the UV sphere space with a continuous radiance field so that an adversarial loss on top of rendered images can supervise the entire pipeline. We achieve high-quality and realistic texture synthesis and substantial improvements over state-of-the-art approaches to texture swapping and editing applications. We are able to generate consistent textures over different object shapes while previous approaches fail. Furthermore, we can generate more diverse textures with the same object shape compared to previous state-of-the-arts.

Despite its merits, our method has inherent limitations. Our current correspondence assumes one-to-one dense mapping. However, this assumption does not always hold in real-world scenarios. See Appendix T for more discussion regarding the limitations. To further achieve more photorealistic textures, one option is to incorporate advanced data-driven priors, such as diffusion models, which can help mitigate the distortions and improve the quality of the generated textures. Utilizing more sophisticated neural rendering architectures, such as ray transformers, can also enhance the results.

## 6 ACKNOWLEDGEMENT

This work was supported, in part, by NSF CAREER Award IIS-2240014, the Amazon Research Award, and the CISCO Faculty Award.

## 7 ETHICS STATEMENT

Our work follows the General Ethical Principles listed in ICLR Code of Ethics (https://iclr.cc/public/CodeOfEthics).

We share similar concerns as other generative models (Poole et al., 2023). Our work may raise ethical concerns related to the improper use of content manipulation or the generation of deceptive information.

Our data generation pipeline relies on a pre-trained diffusion (Rombach et al., 2022) model, which can inherit biases in its training data. The dataset (Schuhmann et al., 2021) used to train StableDiffusion may include undesirable images. Additionally, the pre-trained diffusion model is conditioned on features from another pre-trained large language model, which could introduce unintended biases. As a result, our generated texture may also include unintended biases.

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

# Appendix Table of Contents

## A    MORE QUALITATIVE RESULTS OF CANONICAL SURFACE AUTO-ENCODER

**Can our Canonical Surface Auto-encoder handle different topologies?**    We align with prior works by evaluating our approach on two categories within ShapeNet (e.g., cars, chairs) for comparisons. While using a UV sphere adds constraints on representing shape varieties, our method can still model reasonably diverse topologies. We additionally include the canonical surface auto-encoding results on three different shape collections in Figure 7. Specifically, we include results on ShapeNet (Chang et al., 2015) Airplanes (top), DFAUST (Bogo et al., 2017) (middle, scanned human subjects and motions), and SMAL Zuffi et al. (2017) (bottom, articulated animals e.g., lions, tigers, and horses). Our method consistently produces high-quality shape reconstructions with dense and smooth learned correspondence, even for non-genus zero shapes such as airplanes with holes. This is not achievable for traditional mesh-based UV deformation.

ShapeNet
*(airplanes)*

D-FAUST
*(dynamic human scans)*

SMAL
*(articulated animals)*

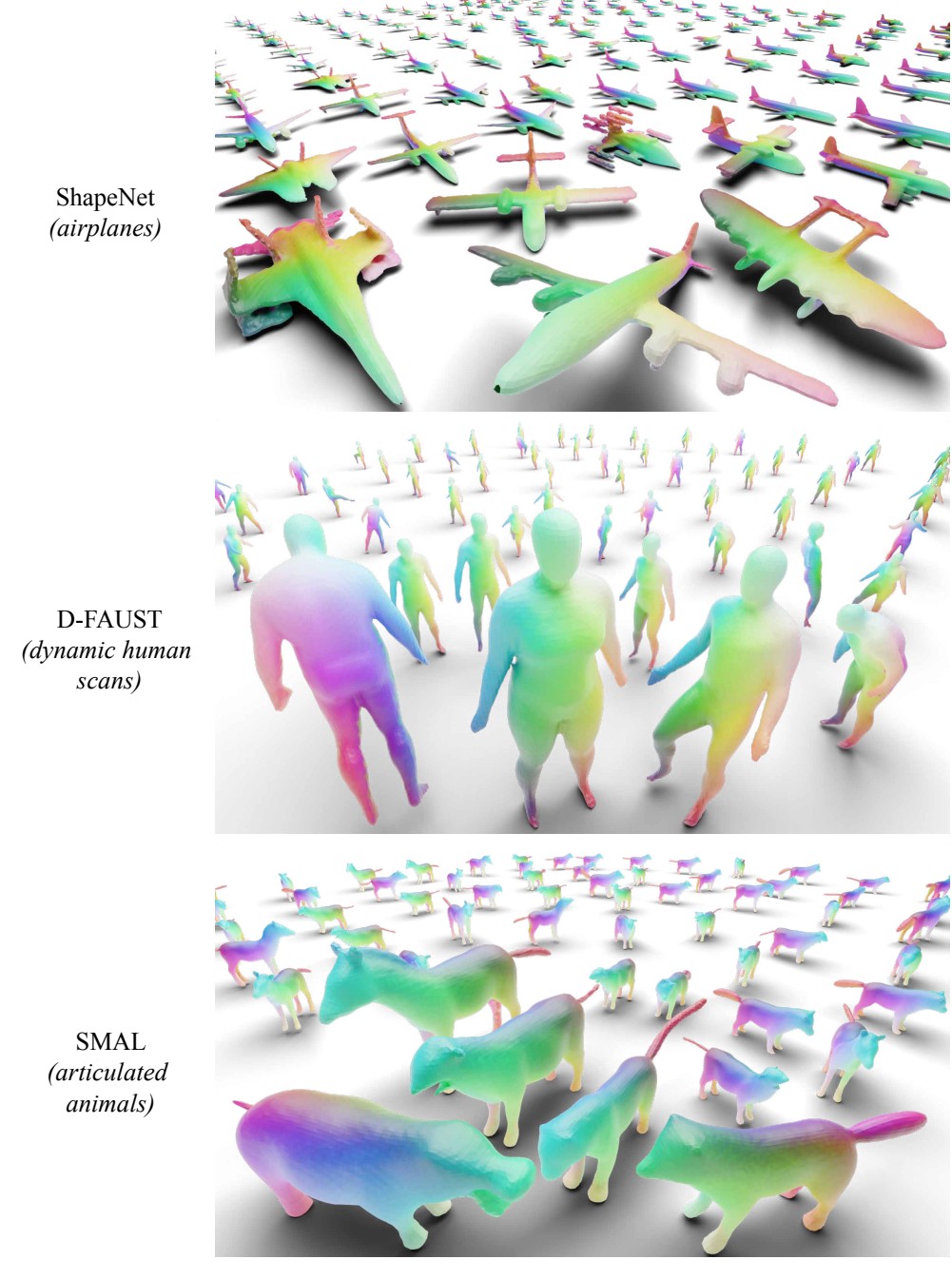

Figure 7: **Surface Reconstruction Results on Different Datasets.**

## B    MORE QUALITATIVE RESULTS USING OUR DATA GENERATION PIPELINE

We provide our texture synthesis results using Canonical Surface Auto-encoder trained on SMAL (Appendix A) and 2D datasets generated by our data pipeline(Appendix P). As shown in Figure 8, we demonstrate that our model can generate photorealistic textures for various animal categories. With the pipeline, we can also control the style with different prompts. Note that for simplicity, some results in the figure are trained on a single instance or view, indicated by grey text.

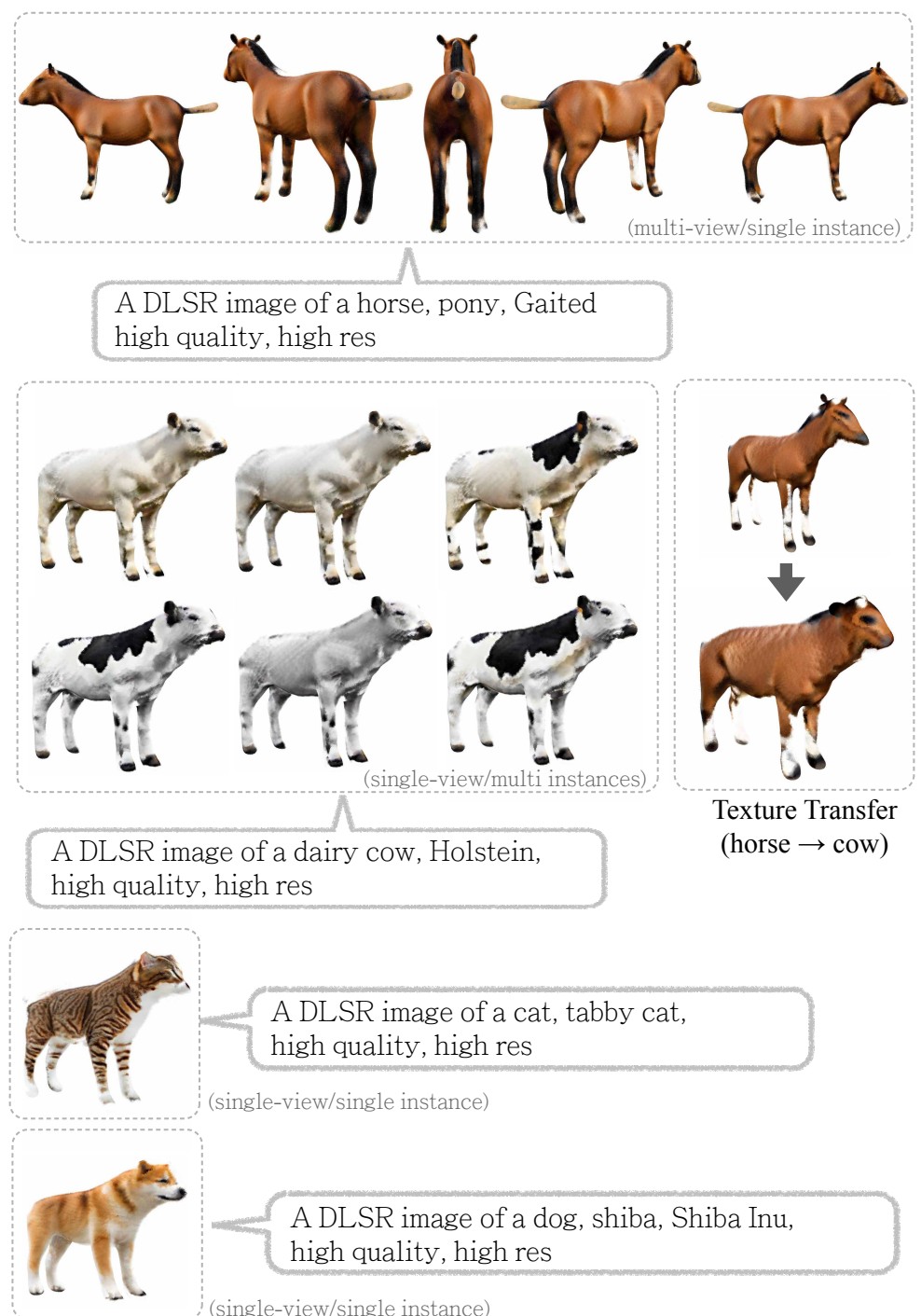

Figure 8: **Texture Synthesis Results on SMAL.** We include results using horses, cows, cats, and dogs. We also transfer a texture from a horse to a cow using correspondence learned from Appendix A.

# C    MORE QUALITATIVE RESULTS OF TEXTURE EDITING

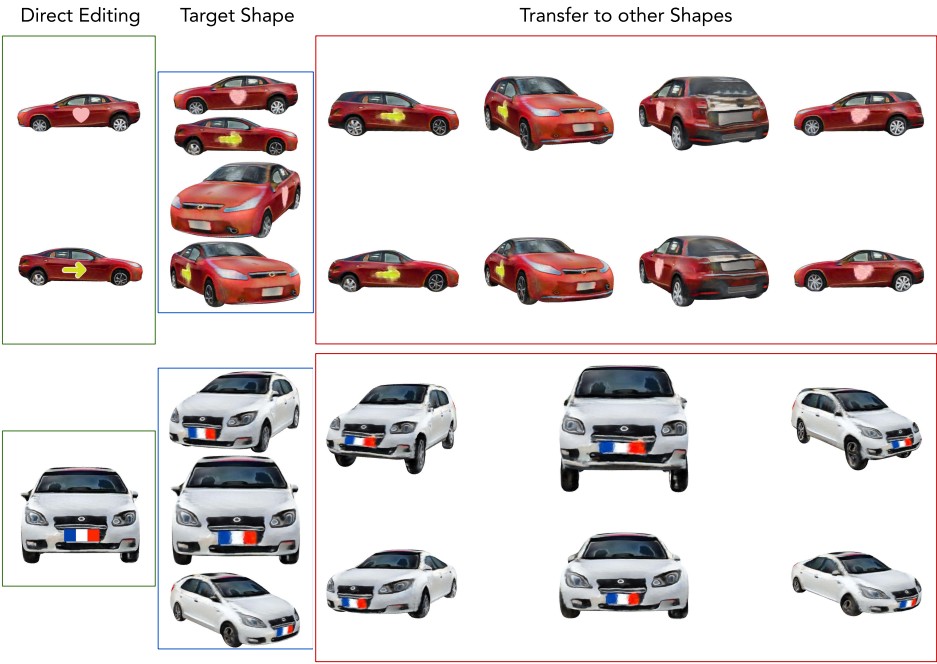

Figure 9: **Texture Editing and Transfer.** Our approach offers exceptional flexibility when it comes to texture editing. We support a range of texture editing techniques, including texture swapping, filling, and painting operations. Given a synthesized texture, one can directly operate on the rendered view to edit the texture (as illustrated by the *green box* ). By fine-tuning the edited image using the back-propagation to the texture feature, we can obtain an edited texture that is 3D consistent across different views (as shown in the *blue box*). Moreover, this edited texture feature can also be transferred among different shapes (as demonstrated by the *red box*).

# D    MORE QUALITATIVE RESULTS OF HIGH-RESOLUTION SYNTHESIS

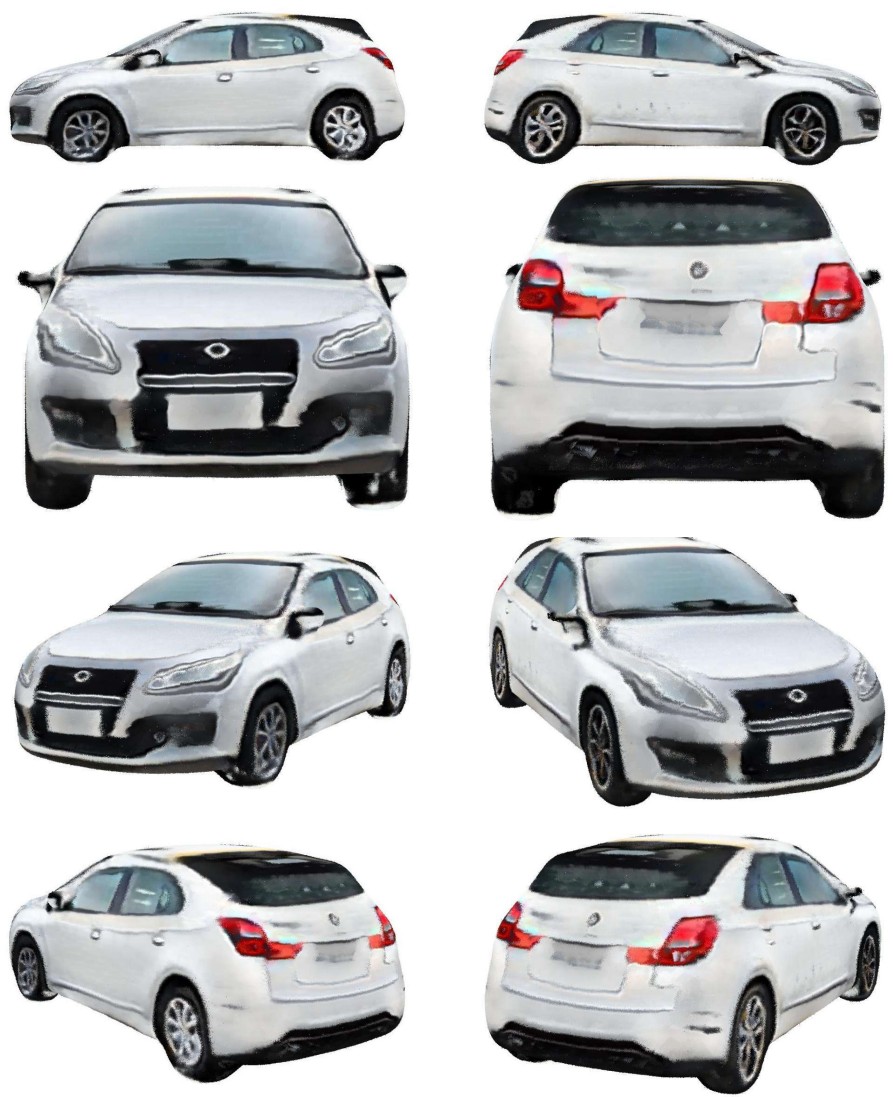

Figure 10: **Our results on Compcars dataset.** The model is trained with $512 \times 512$ resolution, images shown are rendered with $1024 \times 1024$ resolution. Compared to Texturify Siddiqui et al. (2022) (results shown in Figure 13 and Figure 14), our texture synthesis approach produces textures with superior detail. Notably, our generator is capable of synthesizing intricate features such as *logos*, *door handles*, *car wipers*, and *wheel frames*. Zoom in for the best viewing.

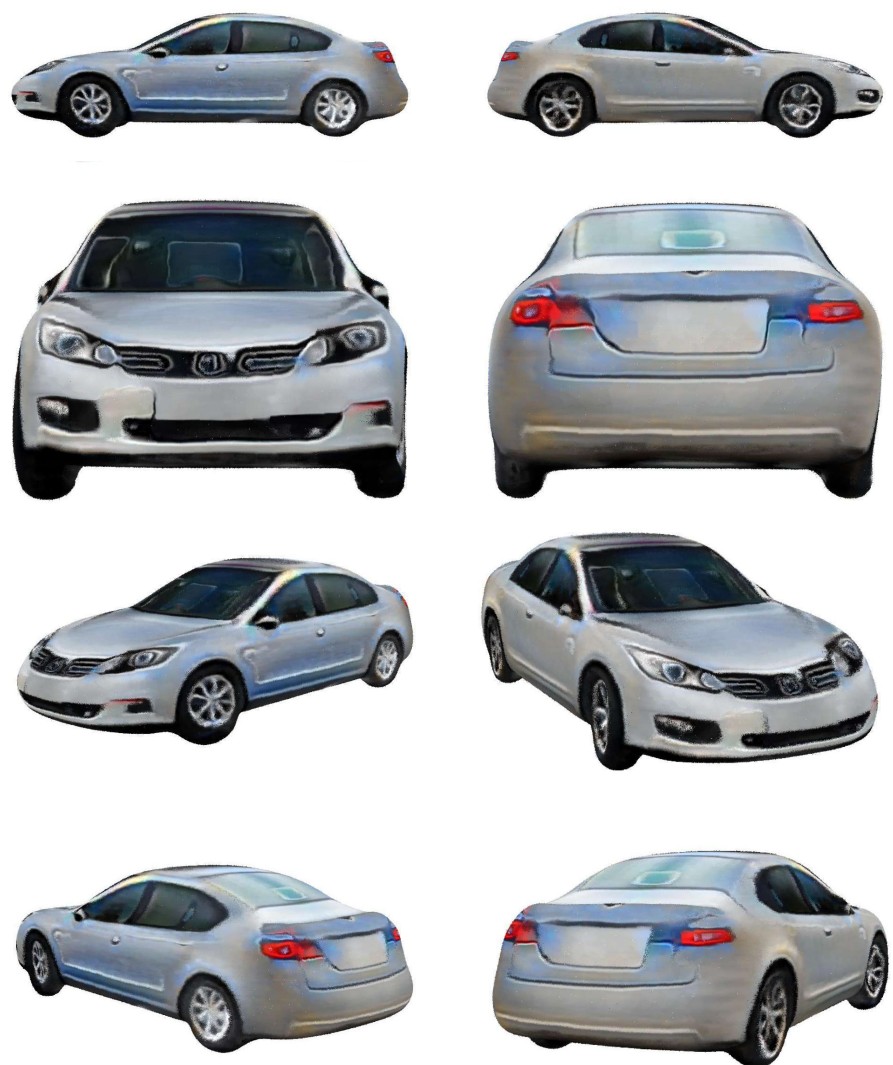

Figure 11: **Our results on Compcars dataset.** The model is trained with $512 \times 512$ resolution, images shown are rendered with $1024 \times 1024$ resolution. Note that all the images are rendered from the same instance, including images in Figure 10 and Figure 12. This highlights the effectiveness of our proposed method in synthesizing photo-realistic textures while maintaining 3D consistency. Zoom in for the best viewing.

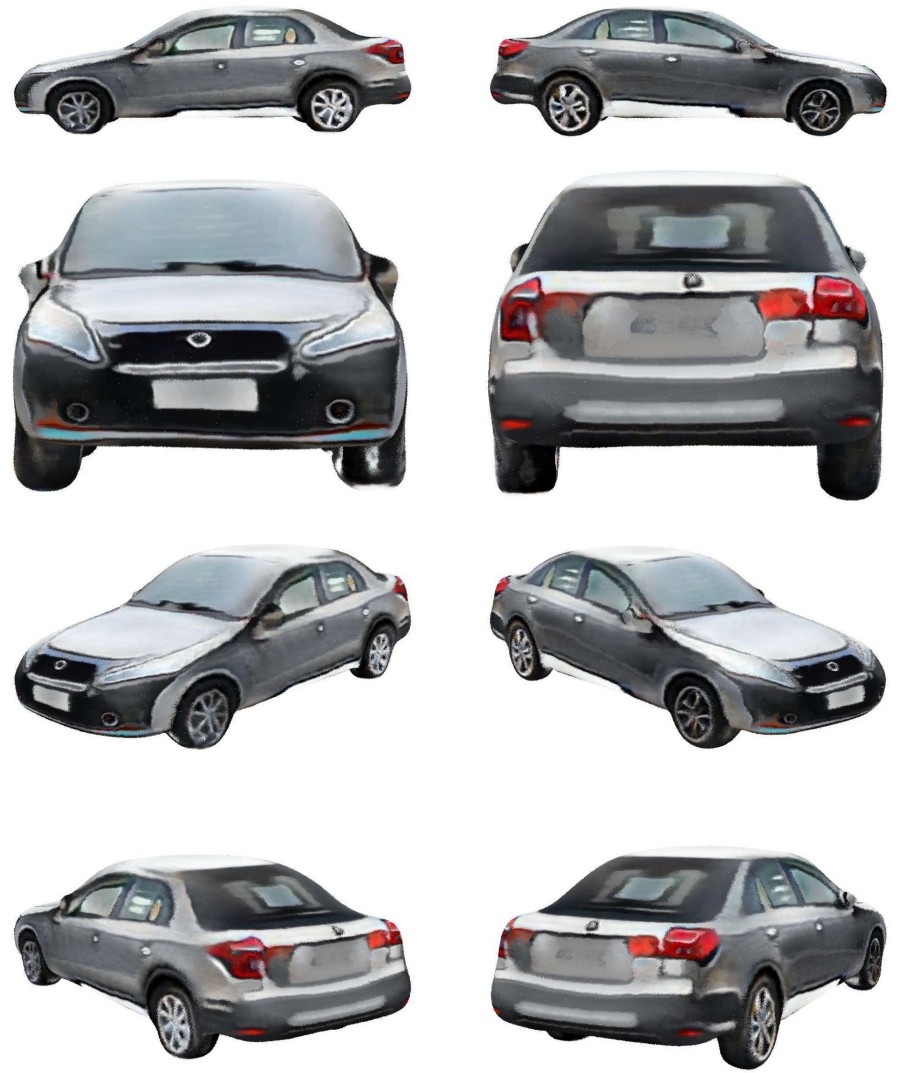

Figure 12: **Our results on Compcars dataset.** The model is trained with $512 \times 512$ resolution, images shown are rendered with $1024 \times 1024$ resolution. In addition to generating different global colors, our proposed method can generate diverse textures by including intricate local details. For example, the generated textures may include *unique logos* (different from those shown in Figure 11) or distinct *tail light styles* (different from Figure 10). Zoom in for the best viewing.

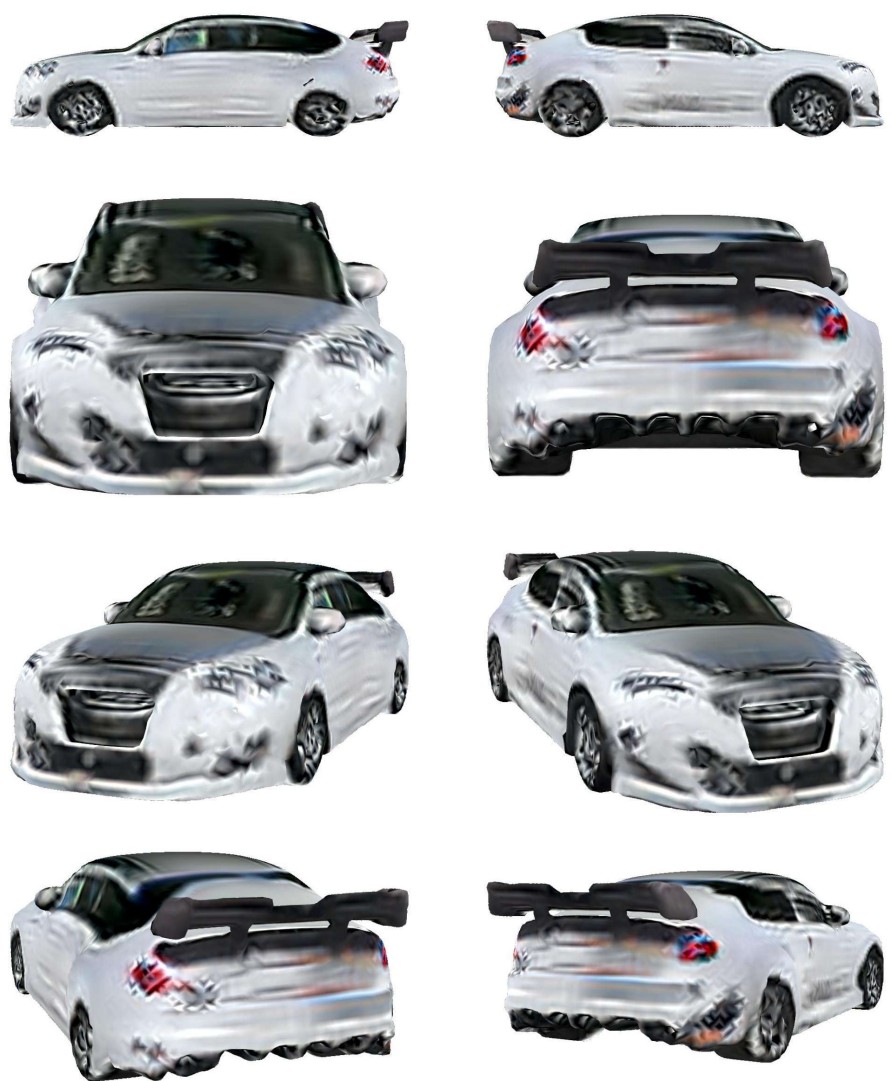

Figure 13: **Texturify** Siddiqui et al. (**2022**) **results on Compcars dataset.** The model is trained with 512×512 resolution, images shown are rendered with 1024×1024 resolution. The sample shown in this figure was generated using the pre-trained model provided by the authors. Notably, all the images in this figure depict different render angles of the same instance.

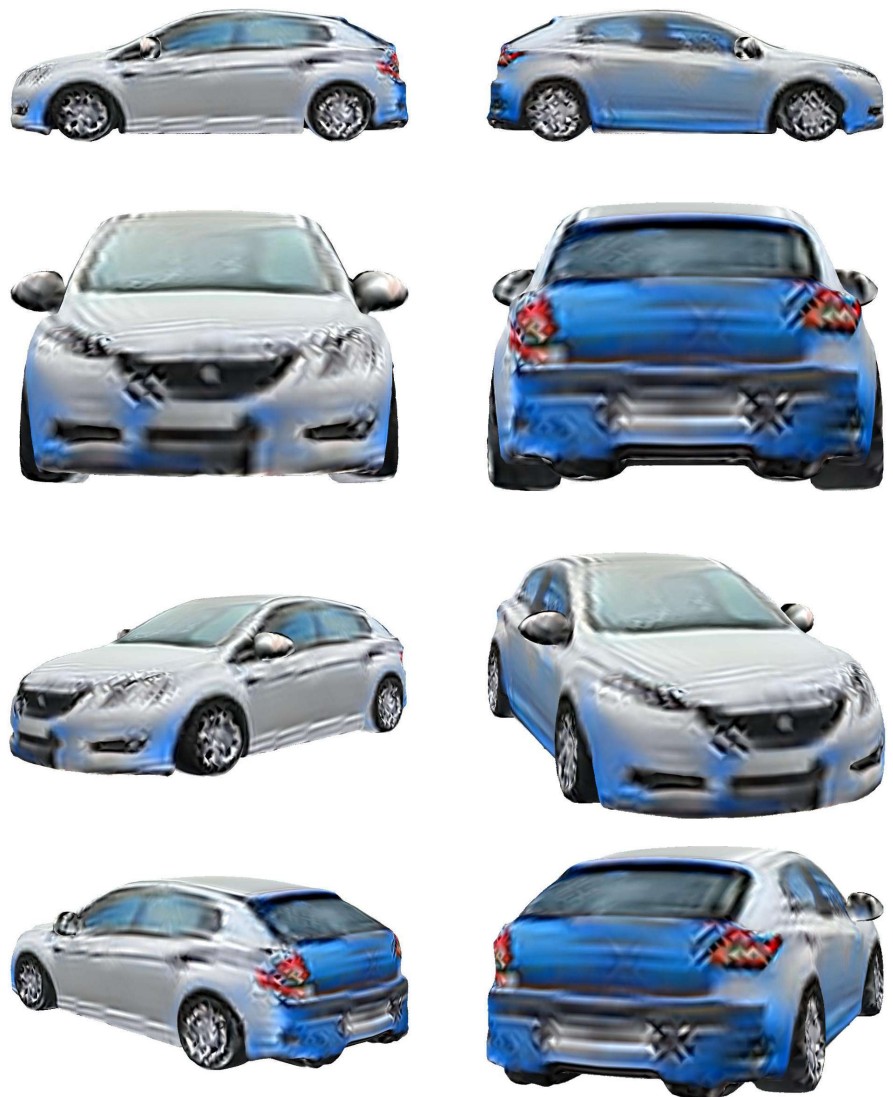

Figure 14: **Texturify Siddiqui et al. (2022) results on Compcars dataset.** The model is trained with 512×512 resolution, images shown are rendered with 1024×1024 resolution. The sample shown in this figure was generated using the pre-trained model provided by the authors. Notably, all the images in this figure depict different render angles of the same instance.

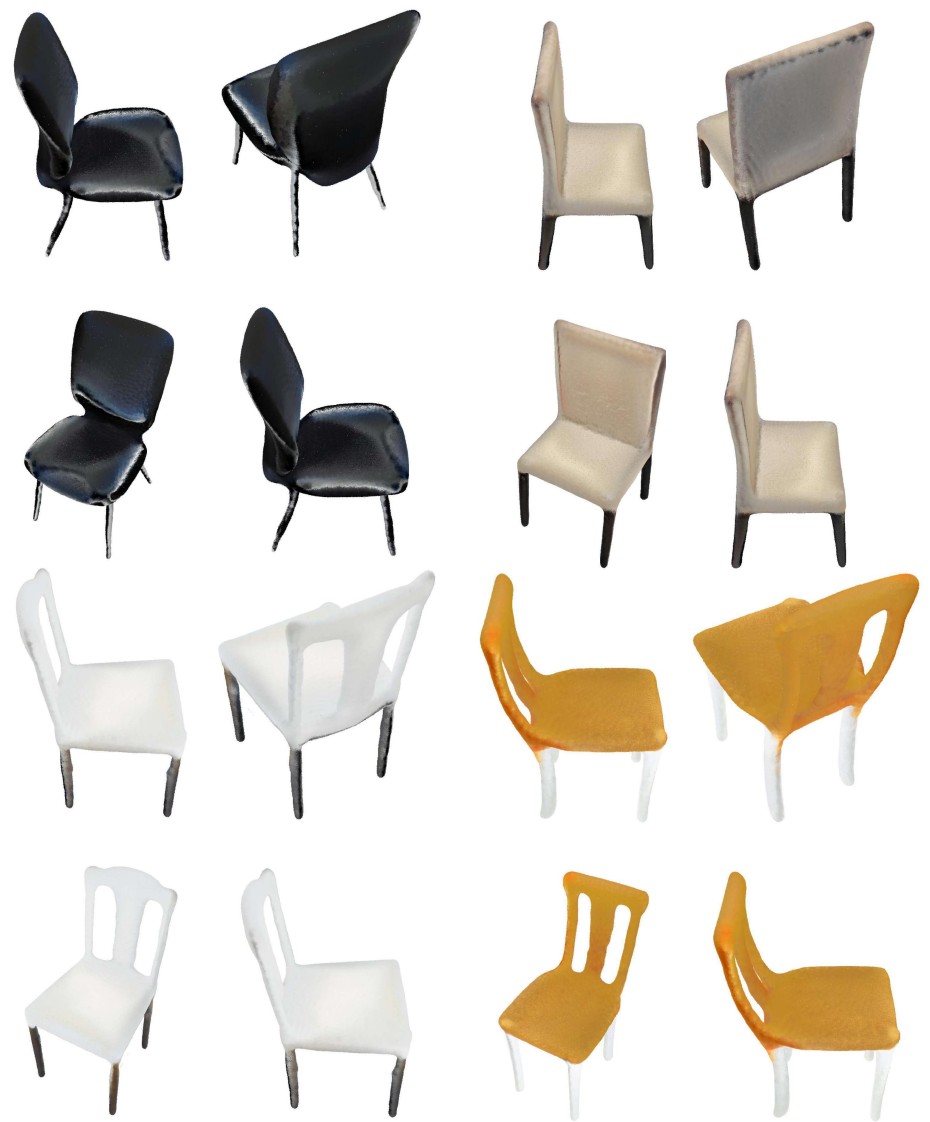

Figure 15: **Our results on Photoshape dataset.** The model is trained with $512\times512$ resolution, images shown are rendered with $1024\times1024$ resolution. Our model is highly effective in synthesizing top-quality textures for chairs. Interestingly, the generated textures may even feature a variety of material styles, such as *black leather*, *suede fabric*, or *flannel* (see Figure 16), adding an extra level of realism to the textures. Zoom in for the best viewing.

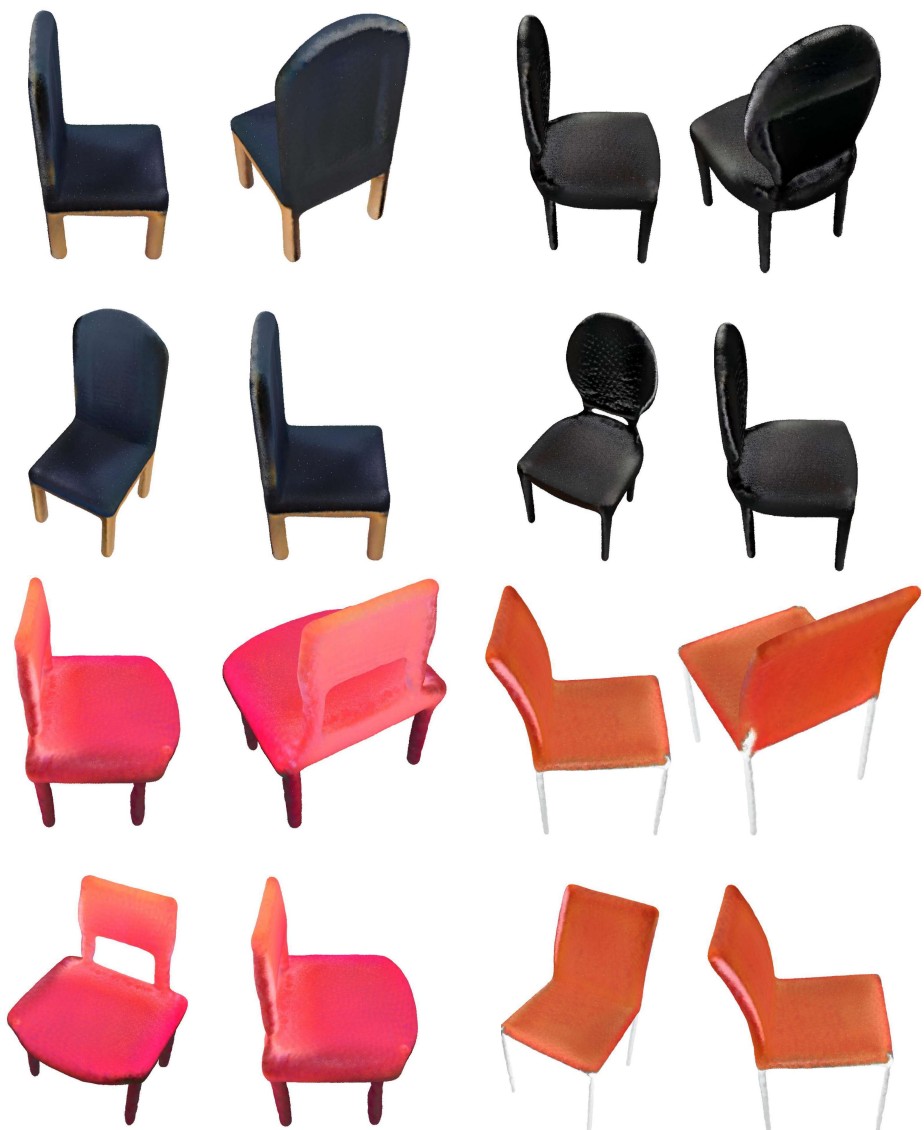

Figure 16: **Our results on Photoshape dataset.** The model is trained with $512 \times 512$ resolution, images shown are rendered with $1024 \times 1024$ resolution. Thanks to the correspondence learned from our Canonical Surface Auto-encoder, textures can be generated without interference from geometry information. Furthermore, the model can predict accurate textures for different parts of the object. For instance, the legs of the chair may have distinct textures from the seats, and the boundary between these two parts is clearly defined. This demonstrates the importance of the correspondence learned from the Canonical Surface Auto-encoder. Zoom in for the best viewing.

# E   MORE QUALITATIVE RESULTS OF TEXTURE TRANSFER

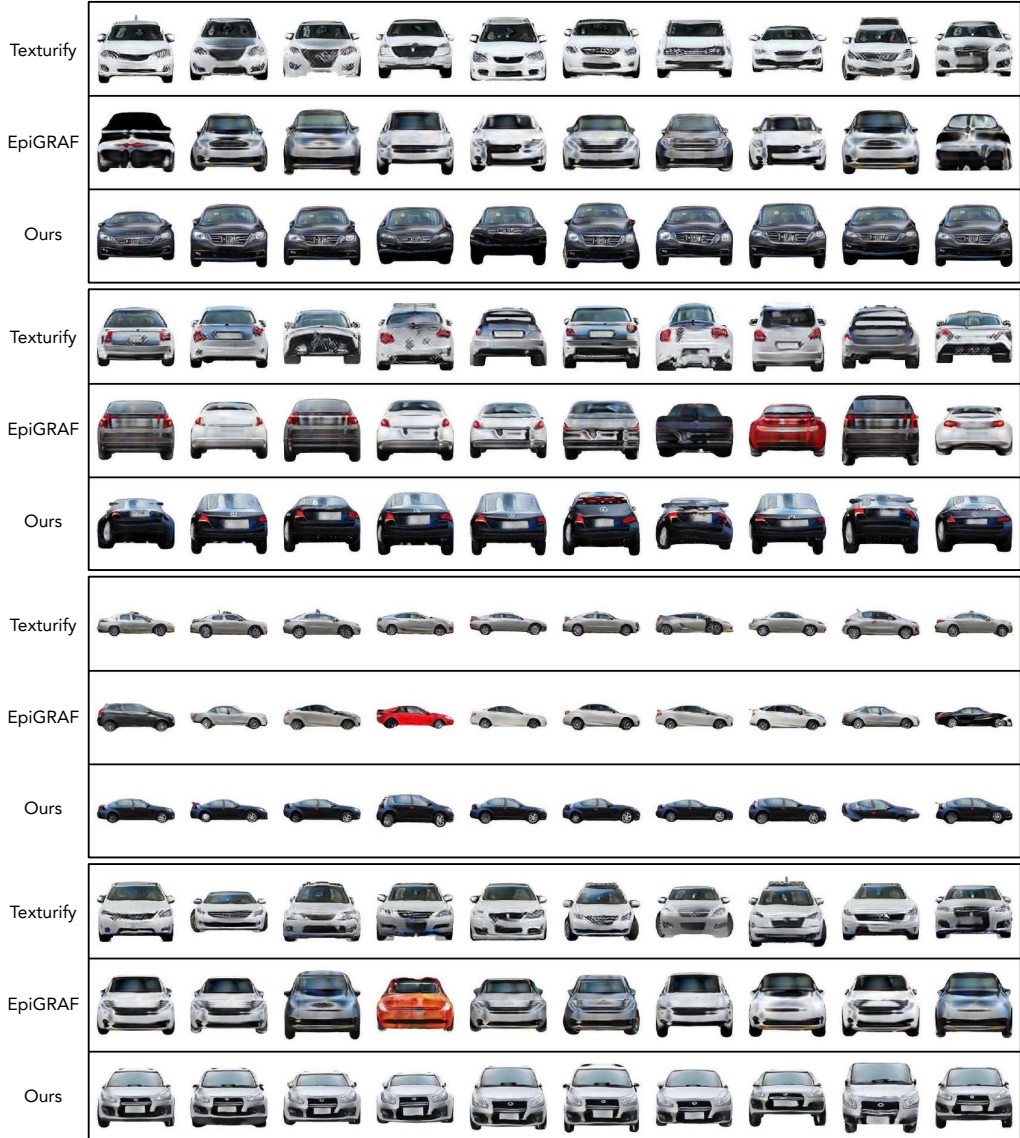

Figure 17: **Texture Transfer Results on CompCars dataset.** Each approach (in the same row) applies the same texture code to synthesize textures on different input shapes. Our method can generate textures that exhibit consistency across all shapes, unlike other approaches (e.g., Texturify Siddiqui et al. (2022) and EpiGRAF Skorokhodov et al. (2022)), which may produce different styles or local details on different object shapes even when using the same texture code.

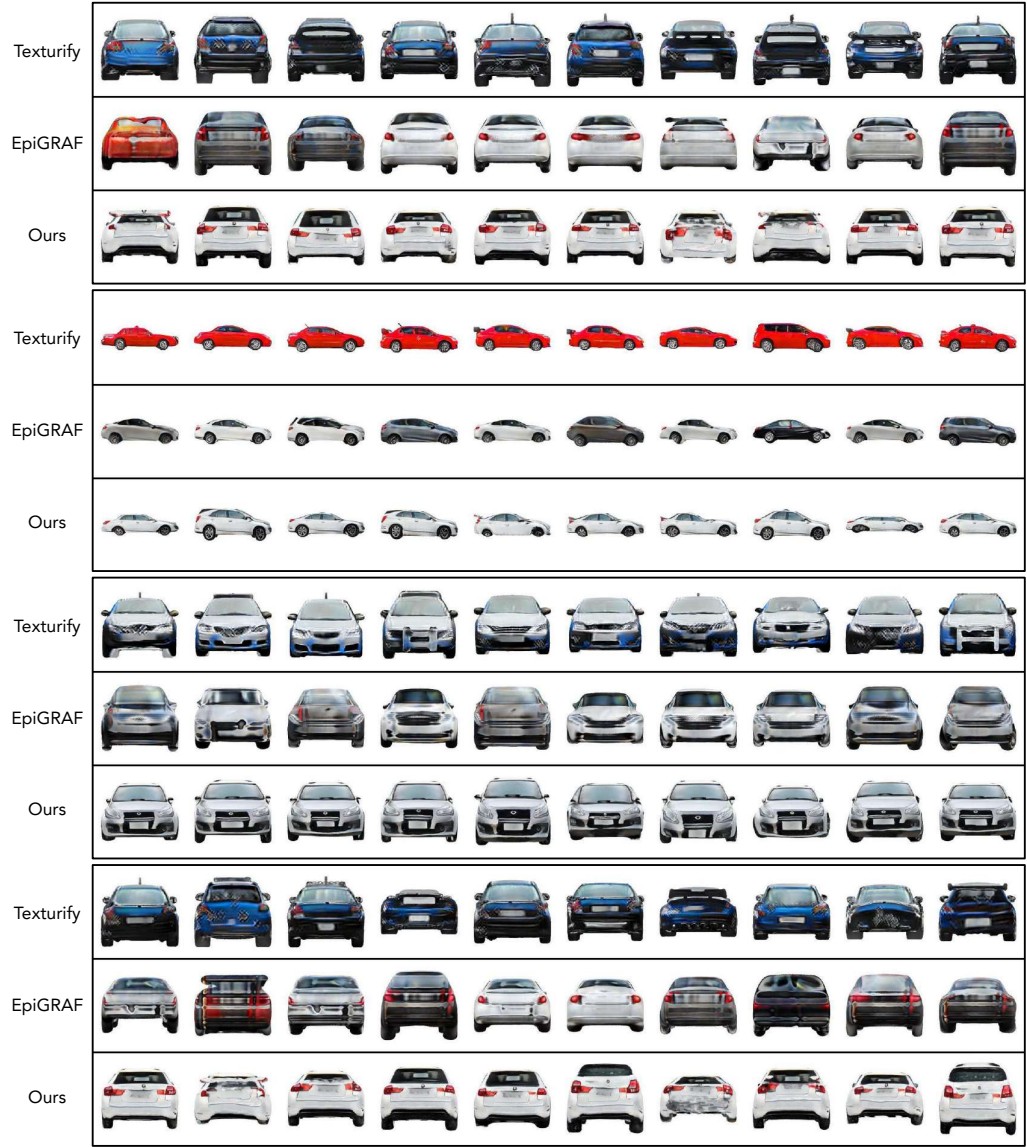

Figure 18: **Texture Transfer Results on CompCars dataset.** Each approach (in the same row) applies the same texture code to synthesize textures on different input shapes. Our method can generate textures that exhibit consistency across all shapes, unlike other approaches (e.g., Texturify Siddiqui et al. (2022) and EpiGRAF Skorokhodov et al. (2022)), which may produce different styles or local details on different object shapes even when using the same texture code. Consider the results shown in row 4 of the figure. While the samples generated by the Texturify Siddiqui et al. (2022) method exhibit consistency in global color (i.e., all the cars are red), the same texture code may result in different window styles (i.e., number of windows).

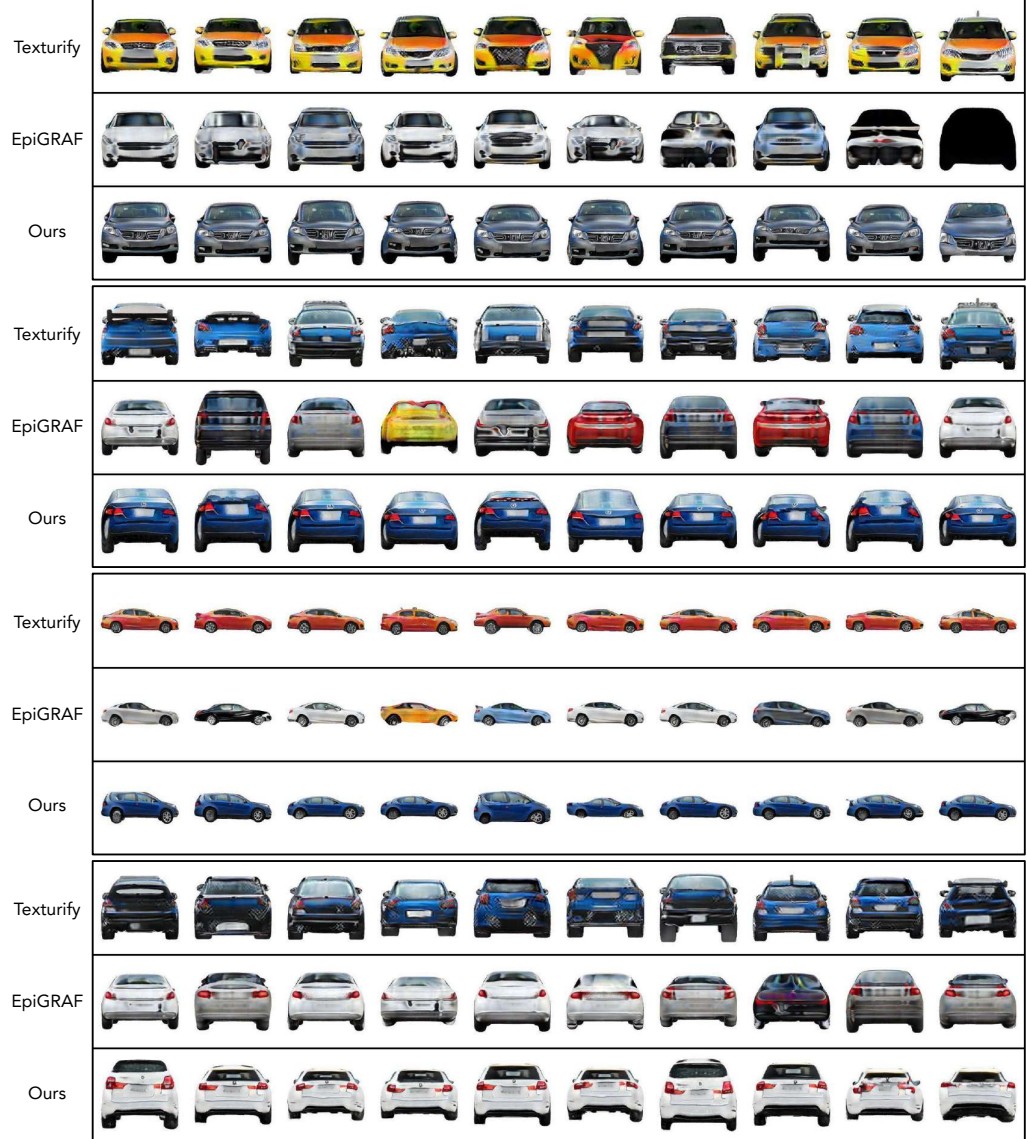

Figure 19: **Texture Transfer Results on CompCars dataset.** Each approach (in the same row) applies the same texture code to synthesize textures on different input shapes. Our method can generate textures that exhibit consistency across all shapes, unlike other approaches (e.g., Texturify Siddiqui et al. (2022) and EpiGRAF Skorokhodov et al. (2022)), which may produce different styles or local details on different object shapes even when using the same texture code.

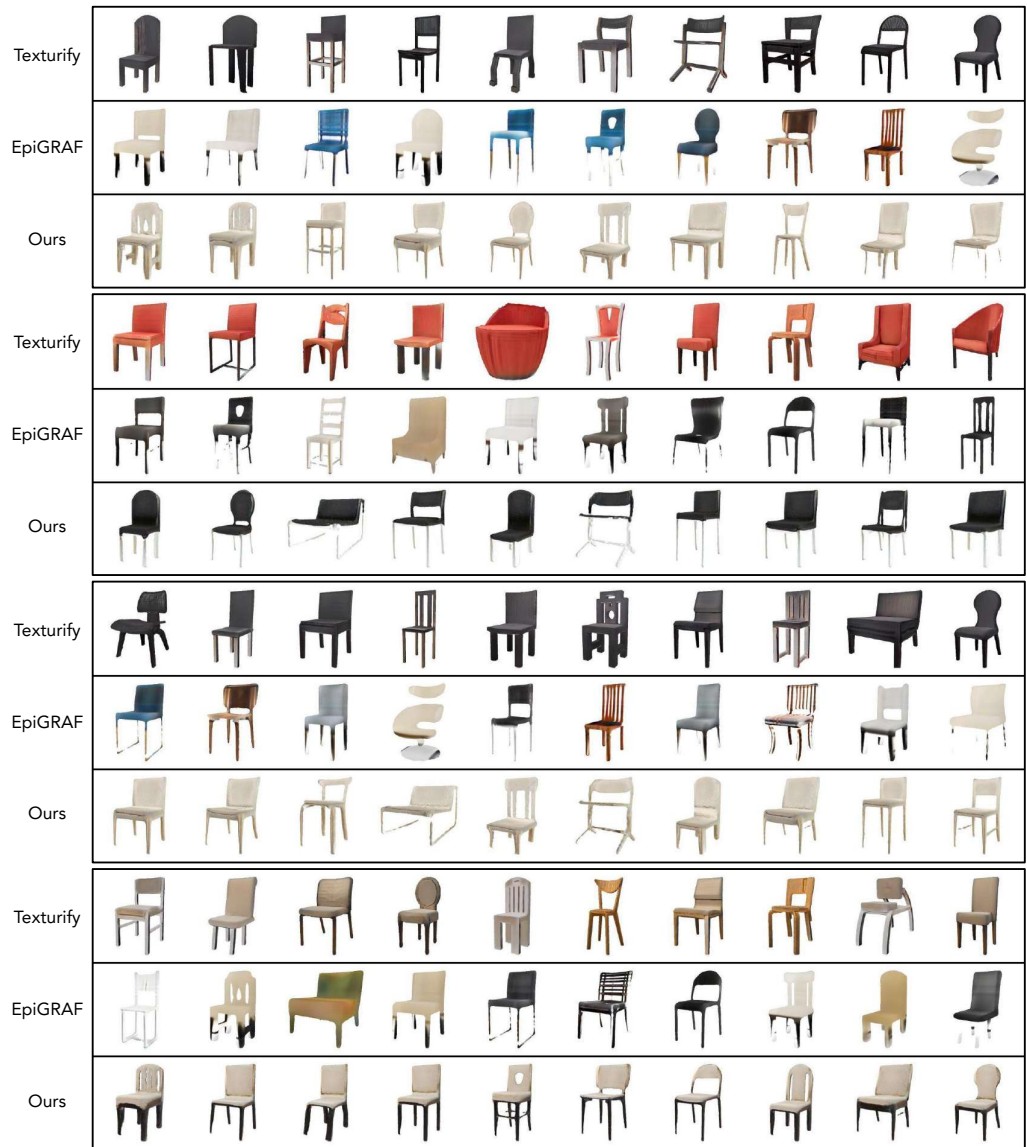

Figure 20: **Texture Transfer Results on Photoshape dataset.** Each approach (in the same row) applies the same texture code to synthesize textures on different input shapes. Our method can generate textures that exhibit consistency across all shapes, unlike other approaches (e.g., Texturify Siddiqui et al. (2022) and EpiGRAF Skorokhodov et al. (2022)), which may produce different styles or local details on different object shapes even when using the same texture code.

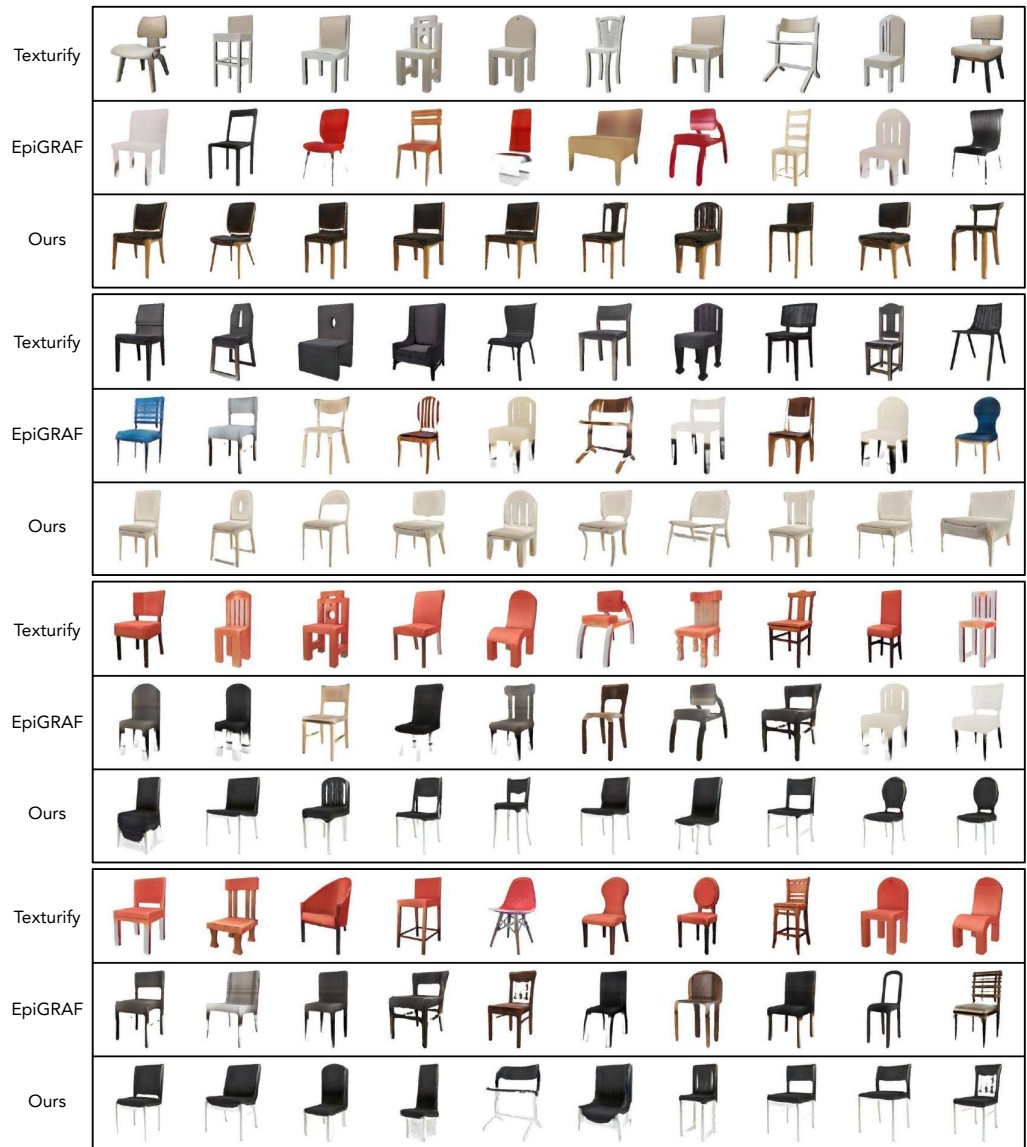

Figure 21: **Texture Transfer Results on Photoshape dataset.** Each approach (in the same row) applies the same texture code to synthesize textures on different input shapes. Our method can generate textures that exhibit consistency across all shapes, unlike other approaches (e.g., Texturify Siddiqui et al. (2022) and EpiGRAF Skorokhodov et al. (2022)), which may produce different styles or local details on different object shapes even when using the same texture code.

## F    MORE QUANTITATIVE RESULTS ON FULL PHOTOSHAPE

We report results on the *full* Phtoshape dataset in Table 5, showcasing superior controllable synthesis but higher FID and KID values compared to Texturify. Similar to prior works (Cheng et al., 2021; 2022), our method builds upon learnable UV maps, assuming one-to-one dense correspondence between instances of a category. However, in real-world scenarios, this assumption does not always hold: There may be variations in shape (e.g., armchairs and straight chairs) or structure (e.g., lounges and bean chairs) across different instances. This introduces challenges in modeling high-fidelity shapes and detailed correspondences at the same time. Despite these challenges, our method produces reasonable results, as depicted in Figure 22.

Table 5: **Quanitative Results on Full Photoshape.** While our method has slightly larger FID and KID than Texturify on full Photoshape, we achieve significantly better results in controllable synthesis. KID is multiplied by $10^2$.

| Method | FID $\downarrow$ | KID $\downarrow$ | LPIPS$_g$ $\uparrow$ | LPIPS$_t$ $\downarrow$ |
|---|---|---|---|---|
| EpiGRAF (Skorokhodov et al., 2022) | 89.74 | 6.28 | 3.63 | 6.51 |
| Texturify (Siddiqui et al., 2022) | 26.17 | 1.54 | 8.86 | 3.46 |
| TUVF (ours) | 57.56 | 3.74 | 16.43 | 2.72 |

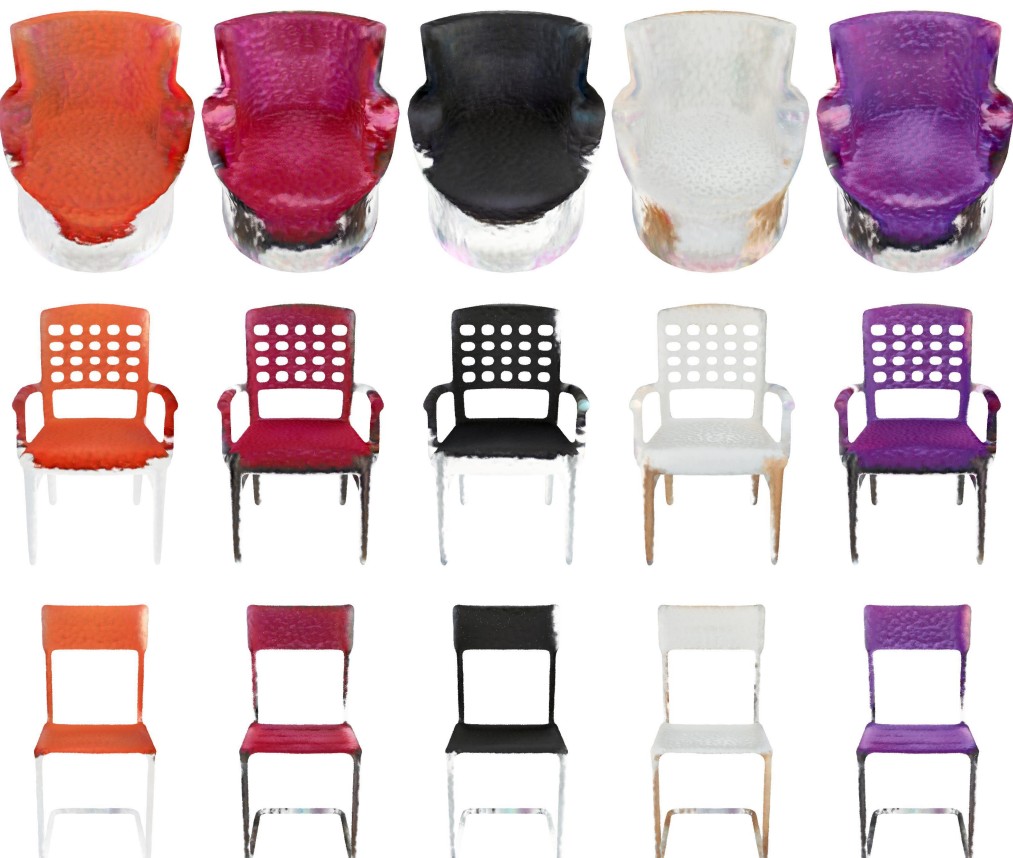

Figure 22: **Qualatative Results on Full Photoshape.** Although finding correspondence for shapes with large structural differences is challenging, our method produces reasonable results.

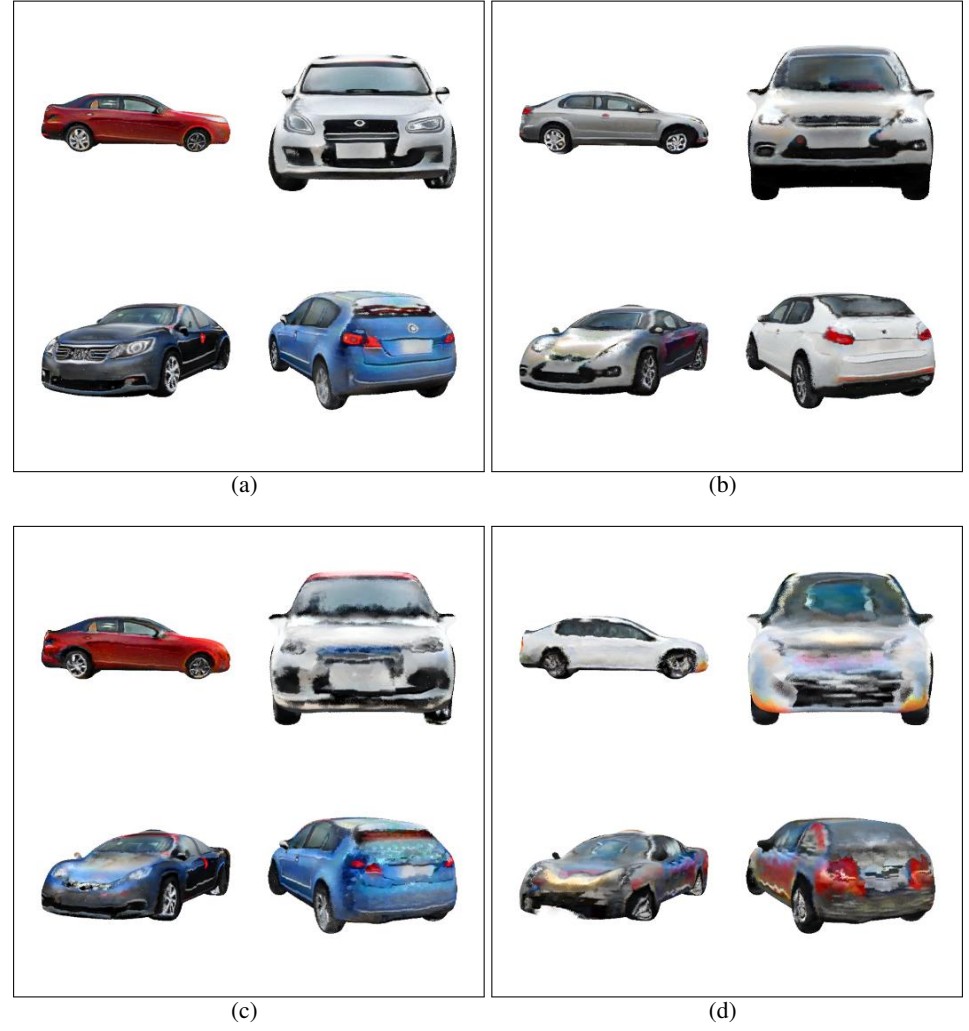

Figure 23: **Visualization of the ablation study over the Canonical Surface Auto-encoder.** The four sub-figures correspond to the four settings introduced in Section H. Zoom in for the best viewing.

## G   MORE COMPARISONS WITH EPIGRAF USING GROUND TRUTH GEOMETRY

For a fair comparison, we use the auto-encoded geometry as the input for both EpiGRAF (Skorokhodov et al., 2022) and our method. This ensures that both approaches utilize the same geometry. Below, we provide the results comparing our method to EpiGRAF while employing the ground-truth SDF for EpiGRAF.

Table 6: **Comparisons with EpiGRAF using Groundtruth Geometry.** KID is multiplied by $10^2$.

| Dataset | Method | FID $\downarrow$ | KID $\downarrow$ | LPIPS$_g$ $\uparrow$ | LPIPS$_t$ $\downarrow$ |
|---|---|---|---|---|---|
| CompCars | EpiGRAF (Skorokhodov et al., 2022) | 88.37 | 6.46 | 4.23 | 2.31 |
| | TUVF (ours) | 41.79 | 2.95 | 15.87 | 1.95 |
| Photoshape | EpiGRAF (Skorokhodov et al., 2022) | 55.31 | 3.23 | 7.34 | 2.61 |
| | TUVF (ours) | 51.29 | 2.98 | 14.93 | 2.55 |

Table 7: **Ablation Study over Canonical Surface Auto-encoder.** Evaluated on CompCars.

| Method | Mapping Direction | GT Geometry | Proxyless Surface | Smootheness | FID↓ | KID↓ |
|---|---|---|---|---|---|---|
| (a) *(base)* | UV → Surface | | ✓ | ✓ | 41.79 | 2.95 |
| (b) | UV → Surface | ✓ | | ✓ | 61.81 | 4.08 |
| (c) | UV → Surface | ✓ | ✓ | | 79.43 | 6.16 |
| (d) | Surface → UV | ✓ | ✓ | | 139.19 | 12.92 |

## H    ABLATION STUDY ON CANONICAL SURFACE AUTO-ENCODER

One drawback of our framework is that the auto-encoded indicator grid may not be perfect. As a result, we investigated several different network designs for the stage-1 geometry pre-training, which enabled us to learn texture synthesis using the ground-truth indicator function. We considered comparing four settings in this study:

(a) *Our Canonical Surface Auto-encoder.* The geometry network takes UV points as inputs and maps them to the surface. An additional function $g_\theta$ is learned to predict the surface normal for each point, and an auto-encoded indicator function is obtained. Texture synthesis is performed using the auto-encoded indicator function.

(b) The geometry network takes UV points as inputs and maps them to the surface. No $g_\theta$ is used. Texture synthesis is performed using the ground-truth indicator function.

(c) The geometry network takes UV points as inputs and maps them to the surface. No $g_\theta$ is used. Texture synthesis uses the ground-truth indicator function, while points are warped to the ground-truth surface via the nearest neighbor.

(d) The geometry network takes surface points as inputs and maps them to the UV. In this case, there is no need for $g_\theta$, and texture synthesis is learned using the ground-truth indicator function.

Two important factors may affect the quality of synthesis. First, the surface points should lie as close to the exact surface of the indicator function as possible. This is because our $\text{MLP}_F$ takes the nearest neighbor feature and the distance between the query point and the nearest neighbor as inputs. If there is a gap between the points and the surface of the indicator function, it can confuse $\text{MLP}_F$ and harm the performance. Secondly, the surface points should be as smooth as possible, i.e., evenly distributed among the surface. This ensures that each surface point contributes to a similar amount of surface area. The results of our ablation study on the car category can be found in Table 7. We also show samples from each setting in Figure 23. The results obtained for settings without smooth correspondence (e.g., setting 3 and 4) show that the textures are more blurry and tend to have distortions. On the other hand, our method produces sharper details compared to setting 2, which is trained on proxy surface points. This study demonstrates the unique advantage of our Canonical Surface Auto-encoder design, in which we can learn UV-to-surface mapping with smooth correspondence. Therefore, learning an additional function $g_\theta$ to predict point normal and obtain an auto-encoded indicator function is necessary to obtain high-fidelity textures.

## I    ABLATION STUDY ON DIFFERENT UV RESOLUTION

Table 8: Ablation Study over different UV resolution. Evaluated on the CompCars dataset.

| UV Resolution | FID↓ | KID↓ |
|---|---|---|
| 2K *(base)* | 41.79 | 2.95 |
| 1K | 43.65 | 3.01 |

We investigated the effect of UV resolution on the quality of our method. To achieve this, we compared our base method with different numbers of UV resolution (1K and 2K). The results in Table 8 showed that increasing the UV resolution leads to improved performance in terms of

producing higher-quality fine-scale details. However, we found that using level 4 ico-sphere vertices (i.e., 2K points) is sufficient to achieve high-quality results. Further increasing the resolution would result in prohibitively long training times due to the K nearest neighbor search. For example, using level 5 ico-sphere vertices would result in 10242 points, which would significantly slow down the training speed.

## J  IMPLEMENTATION DETAILS OF TUVF RENDERING

To obtain samples from the UV sphere, we use the vertices of a level 4 ico-sphere, which provides us with 2562 coordinates. After passing these coordinates through the mapping functions $f_\theta$, $g_\theta$, and $h_\theta$, we obtain 2562 surface points ($X_{p'}$), 2562 surface normal ($N_{p'}$), a $128^3$ indication function grid ($\chi'$), and 2562 32-dimensional texture feature vectors. To compute the final color of a ray, we first sample 256 shading points and identify the valid points using the indication function grid $\chi'$. Next, we sample three valid shading points around the surface, and for each valid shading location $x_i$, we conduct a K-nearest neighbor search on the surface points $X_{p'}$. We perform spatial interpolation of the texture feature on the K-nearest surface points to obtain the texture feature $c_{x_i}$ for the current shading points $x_i$. In our experiment, we set K to 4, which is not computationally expensive since we only deal with 2562 surface points.

## K  IMPLEMENTATION DETAILS OF CANONICAL SURFACE AUTO-ENCODER

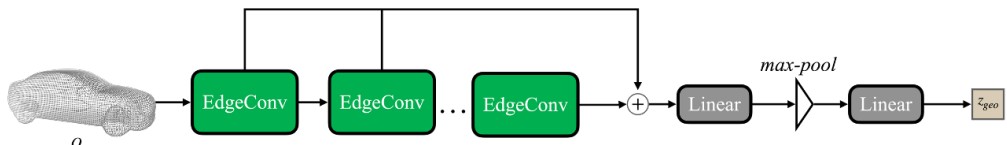

Figure 24: **Architecture of our Shape Encoder $\mathcal{E}$.**

**Shape Encoder $\mathcal{E}$.**  Given a 3D object $O$, we first normalize the object to a unit cube and sample 4096 points on the surface as inputs to the Shape Encoder $\mathcal{E}$ Cheng et al. (2021). The encoder structure is adopted from DGCNN Wang et al. (2019), which contains 3 EdgeConv layers using neighborhood size 20. The output of the encoder is a global shape latent $z_{geo} \in R^d$ where $d = 256$.

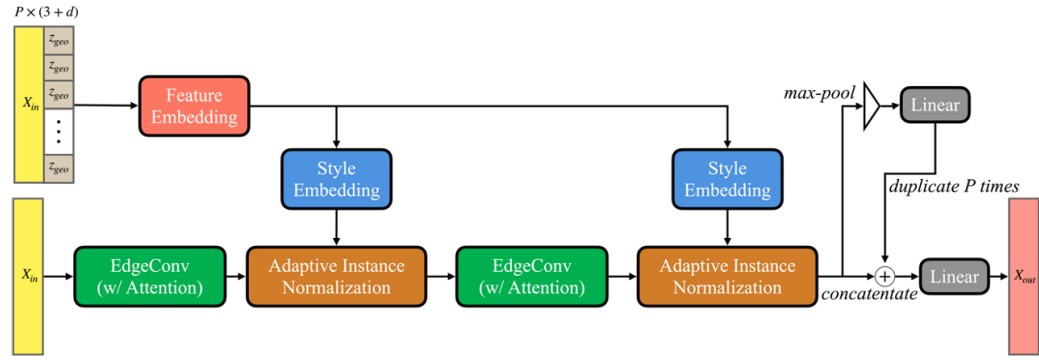

Figure 25: **Architecture of our Surface Points Decoder $f_\theta$ and Surface Normal Decoder $g_\theta$.**

**Surface Points Decoder $f_\theta$ and $g_\theta$.**  Both the surface points decoder $f_\theta$ and surface normal decoder $g_\theta$ share the same decoder architecture, which is adapted from Cheng et al. (2022). We show the detailed architecture of our decoder in Figure 25. The decoder architecture takes a set of point coordinates $X_{in}$ and geometry feature $z_{geo}$ as input and learns to output a set of point coordinates $X_{out}$ in a point-wise manner. To process a set of point coordinates $X_{in}$ and geometry feature $z_{geo}$, the decoder first creates a matrix by duplicating $z_{geo}$ for each coordinate in $X_{in}$ and concatenating it to each coordinate. This matrix includes both the point coordinates and geometry features. The

decoder has two branches: one branch uses an EdgeConv Wang et al. (2019) with an attention module to extract point-wise spatial features from the point coordinates. The attention module is adopted from Li et al. (2021b), which regresses additional weights among the $K$ point neighbors' features as attentions. The other branch employs a nonlinear feature embedding technique to extract style features from the geometry feature. The local styles are then combined with the spatial features using adaptive instance normalization Dumoulin et al. (2016) to create fused features. The style embedding and fusion process is repeated, and finally, the fused feature is used to predict the final output $X_{out}$. It is worth noting that both the surface points decoder $f_\theta$ and surface normal decoder $g_\theta$ use the same geometric features as input. However, they differ in their coordinate input. Specifically, $f_\theta$ takes 2562 UV coordinates as input, while $g_\theta$ uses the 2562 output coordinates from $f_\theta$ as input.

## L    IMPLEMENTATION DETAILS OF TEXTURE GENERATOR $h_\theta$ CIPS-UV

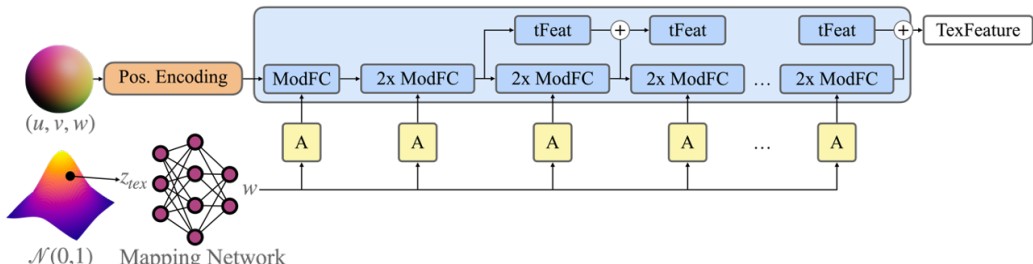

Figure 26: **Architecture of our UV texture feature generator $h_\theta$.** A denotes the Affine Transformation module Karras et al. (2019), ModFC denotes modulated fully connected layers, and tFeat denotes temporary features.

Our generator network, which has a multi-layer perceptron-type architecture Anokhin et al. (2021), is capable of synthesizing texture features on a UV sphere. To achieve this, we use a random texture latent vector $z_{tex}$ that is shared across all UV coordinates, as well as the UV coordinates $(u, v, w)$ as input. The generator then returns the 32-dim texture feature vector value $c$ for that particular UV coordinate. Thus, to compute the entire UV sphere, the generator is evaluated at every pair of coordinates $(u, v, w)$ while keeping the texture latent vector $z_{tex}$ fixed. Specifically, we utilize a mapping network to convert the random texture latent vector $z_{tex}$ into a style vector with the same dimension as $z_{tex}$. This vector injects style into the generation process through weight modulation. We follow the Fourier positional encoding method outlined in Anokhin et al. (2021) to encode the input UV coordinates. The resulting coordinate features pass through the modulated fully connected layers (ModFC), which are controlled by the style vector mentioned above. Finally, we obtain a 32-dimensional texture feature for the input coordinate.

## M    IMPLEMENTATION DETAILS OF PATCH-BASED DISCRIMINATOR

Our discriminator is based on EpiGRAF Skorokhodov et al. (2022), which is similar to the one used in StyleGAN2 Karras et al. (2020b), but modulated by the patch location and scale parameters. We follow the patch-wise optimization approach for training, along with using Beta distribution Skorokhodov et al. (2022) for sampling the scale. We use an initial beta value of $1e^{-4}$ and gradually anneal it to 0.8 after processing $1e^7$ images.

## N    TRAINING DETAILS AND HYPER-PARAMETERS

To demonstrate the training pipeline, we use the car category in ShapeNet and the CompCars dataset as examples. All experiments are performed on a workstation equipped with an AMD EPYC 7542 32-Core Processor (2.90GHz) and 8 Nvidia RTX 3090 TI GPUs (24GB each). We implement our framework using PyTorch 1.10. For further details and training time for each stage, please refer to Algorithm 1.

---

**Algorithm 1 :** The training phase of our approach consists of two stages: (1) Canonical Surface Auto-encoder (2) Texture Feature Generator using adversarial objectives

---

**(A) CANONICAL SURFACE AUTO-ENCODER**  $\triangleright$ *12 hours on ShapeNet Car dataset*

1:  Sub-sample points from the input point clouds as $x$ and the canonical UV sphere $\pi$;
2:  Compute ground-truth indicator function grid $\chi$;
3:  Initialize weights of the encoder $\mathcal{E}$, decoder $f_\theta$ and $g_\theta$;
4:  **while** not converged **do**
5:     **foreach** iteration **do**
6:        $z_{geo} \leftarrow \mathcal{E}(x)$;
7:        $\hat{x} \leftarrow f_\theta([\pi_i, z_{geo}])$, where $\pi_i \in \pi$;
8:        $\hat{n} \leftarrow g_\theta([\hat{x}_i, z_{geo}])$, where $\hat{x}_i \in \hat{x}$;
9:        $\chi' \leftarrow dpsr(\hat{x}, \hat{n})$;
10:      Obtain reconstruction loss $L_{CD}(\hat{x}, x)$ and $L_{DPSR}(\chi', \chi)$;
11:      Update weight;

---

**(B) TEXTURE FEATURE GENERATOR**  $\triangleright$ *36 hours on CompCars dataset*

1:  Sample points from the canonical sphere $\pi$;
2:  Random sample shapes with point cloud $x$ and images from dataset $I_{real}$;
3:  Load pre-trained encoder $\mathcal{E}$, $f_\theta$ and $g_\theta$;
4:  Initialize weights of the texture feature generator $h_\theta$ and patch-based discriminator D;
5:  **while** not converged **do**
6:     **foreach** iteration **do**
7:       Obtain $\hat{x}$, $\hat{n}$, and $\chi'$ with encoder $\mathcal{E}$, $f_\theta$ and $g_\theta$;
8:       Sample $z_{tex}$ from multivariate normal distribution;
9:       $c_i \leftarrow h_\theta(\pi_i, z_{tex})$, where $\pi_i \in \pi$;
10:      $I_{fake} \leftarrow \mathcal{R}(\hat{x}, c, \chi', d)$, where $\mathcal{R}$ denotes renderer and $d$ are camera angles;
11:      Obtain loss $L_{GAN}(I_{fake}, I_{real})$;
12:      Update weight;

---

## N.1 DATA AUGMENTATIONS AND BLUR.

Direct applying the discriminator fails to synthesize reasonable textures since there exists a geometric distribution shift exists bet collection and rendered 2D images. Therefore, following (Chan et al., 2022; Skorokhodov et al., 2022), we apply Adpative Discriminator Augmentation (ADA) (Karras et al., 2020a) to transform both real and fake image crops before they enter the discriminator. Specifically, we use geometric transformations, such as random translation, random scaling, and random anisotropic filtering. However, we disable color transforms in ADA as they harm the generation process and result in undesired textures. In addition to ADA, we also blur the image crops, following (Chan et al., 2022; Skorokhodov et al., 2022). However, since we use larger patch sizes, we employ a stronger initial blur sigma (i.e., 60) and a slower decay schedule, where the image stops blurring after the discriminator has seen $5 \times 10^6$ images.

## O COMPUTATIONAL TIME AND MODEL SIZE

Table 9: **The parameter size and inference time for different models.** Inference time is measured in seconds.

| Method | Representation | Feature Parameterization | Model Size ↓ | Inference Time ↓ |
|---|---|---|---|---|
| Texturify | Mesh | 24K Faces | 52M | 0.2039 |
| EpiGRAF | NeRF | 128×128 Triplanes | 31M | 0.2537 |
| Ours | NeRF | 2K Point Clouds | 9M | 0.3806 |

We provide a comparison of the inference time and model size of different models in Table 9. Specifically, we measure the inference time and size of each model based on the time and number

of parameters required to generate a texture for a given shape instance and render an image of resolution 1024. All experiments are conducted on a workstation with an Intel(R) Core(TM) i7-12700K (5.00GHz) processor and a single NVIDIA RTX 3090 TI GPU (24GB). Texturify is a mesh-based approach and is more efficient in terms of rendering compared to NeRF-based methods. However, its feature space is heavily parameterized on the faces, which makes it memory inefficient. Similarly, EpiGRAF requires computing high-resolution triplanes, making it memory-intensive. In contrast, we only parametrize on 2K point clouds throughout all the experiments and can achieve comparable or even better fidelity. Note that we use the same rendering approach for both TUVF and EpiGRAF; therefore, EpiGRAF has a lower inference time than TUVF because it does not require KNN computation.

## P  IMPLEMENTATION DETAILS OF DATA GENERATION PIPELINE

We utilized Stable Diffusion models Rombach et al. (2022) to generate realistic texture images, which were subsequently used as training data for TUVF. We start by rendering depth maps from synthetic objects using Blender and converting these depth maps into images using depth-conditioned Controlnet Zhang et al. (2023a). If the 3D shape contains object descriptions in its metadata (e.g., ShapeNet (Chang et al., 2015)), we use the description as text prompt guidance. After generating the image, we determine the bounding box based on the depth map and feed this into the Segment Anything Model (SAM) Kirillov et al. (2023) to mask the target object in the foreground. This results in realistic textures for synthetic renders. Our pipeline eliminates the need for perfectly aligned cameras and mitigates differences between 3D synthetic objects and 2D image sets.

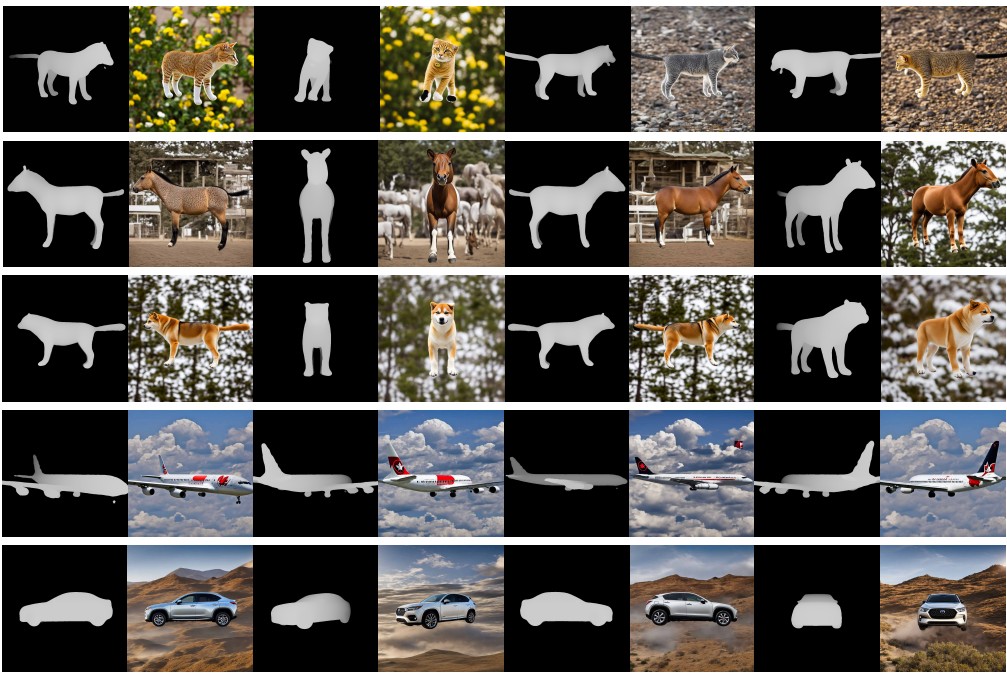

Figure 27: **Examples of 2D Images Generated by our dataset pipeline.** Zoom-in is recommended.

## Q  SAMPLES OF OUR GENERATED DATASET

We show samples of the depth map and its corresponding 2D images that Controlnet generated on four categories (e.g., cats, horses, dogs, airplanes, and cars). Our pipeline can automatically generate realistic and high-quality (1024× 1024) 2D textured images for 3D models. For the DiffusionCat dataset, we use 250 shapes from SMAL, and split them into 200 for training and 50 for testing. We use all 250 shapes to generate textured images. Specifically, we generate 2 samples for eight views for each shape, which results in 4000 images. We use 20 denoise steps for Controlnet, and the entire process takes less than 12 hours for a single Nvidia GeForce RTX 3090.

# R    MOTIVATION FOR USING THE NEURAL RADIANCE FIELD

Our use of NeRF representation is highly motivated and intuitive rather than just a trendy choice. It provides a necessary mechanism for learning surface representation from pixels without any paired (mesh, textured mesh) training data. To understand why we opted for NeRF, we discuss three other options using mesh representation and their limitations. We find that without the help of NeRF, there is no straightforward way to maintain correspondence on the faces while ensuring good geometry.

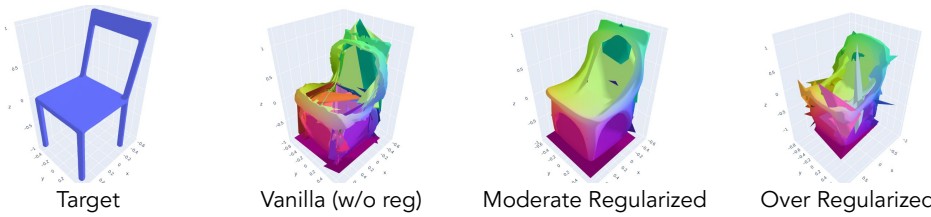

| Target | Vanilla (w/o reg) | Moderate Regularized | Over Regularized |

Figure 28:  **Results using Direct Mesh Deformation.**

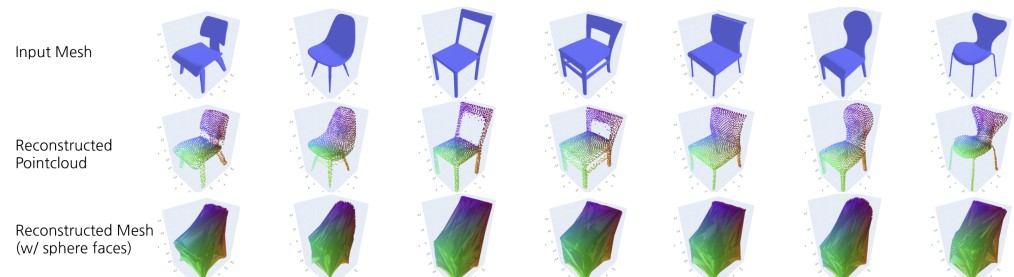

Figure 29:  **Results using Canonical Point Auto-encoder + Sphere Faces.**

1. **Direct Mesh Deformation.** One way to build dense correspondence using mesh representation is through direct deformation. This approach involves deforming a template mesh with the offset of each vertex while maintaining the mesh's topology. We show the results in Figure 28 using direct mesh deformation targeting this chair. The main limitation of this approach is that it struggles with non-genus-zero shapes and tends to produce non-smooth results, even with various shape regularizers. We tested the method with multiple shape regularizers, including $L_{edge}$, $L_{normal}$, and $L_{laplacian}$, with different balancing weights. We also included images of the results when no regularizer was used (noted as Vanilla).

2. **Canonical Point Auto-encoder + Sphere Faces.** Canonical Point Auto-encoder can handle non-genus-zero shapes by allowing complex transformations like folding, cutting, squeezing, and stretching. Therefore, using the same faces as the points on the canonical sphere would lead to incorrect meshes. We show the results in Figure 29.

3. **Canonical Surface Autoencoder + Marching Cubes.** While this method maintains correspondence for points and can reconstruct shapes that are not genus-zero, the faces generated by marching cubes lack correspondence. Therefore, these faces cannot be used for our purpose.

These challenges led us to choose the neural radiance field over the mesh representation for TUVF. The neural radiance field offers a continuous representation that aligns naturally with our design goals. Given that our canonical point auto-encoder already provides continuity by mapping every point on the sphere to the shape's surface, using a continuous representation is a logical and intuitive choice. To further ensure the continuity, we introduce an implicit CIPS-UV generator to achieve high-resolution and high-fidelity output.

## S  DISSCUSION ON MORE RELATED WORK

Texture synthesis is a popular field, with several recent works dedicated to it. While they all aim to synthesize high-quality textures for given shapes, they each have different setups. Some prioritize per-object optimization, some lack correspondence capabilities and others demand 3D supervision. TUVF stands out by focusing on learning generalizable, disentangled, and realistic textures for untextured collections of 3D objects in an unsupervised manner (without paired training data for supervised learning). What sets TUVF apart is its reliance on easily accessible images and untextured shape datasets, eliminating the need for explicit 3D color supervision. This is crucial because, to our knowledge, no large-scale realistic textured 3D dataset is currently available. Even recent efforts like Objaverse (Deitke et al., 2023), while providing substantial 3D data, have limited inter-class sample numbers (in the hundreds), and most of the textures are not photorealistic (compared to natural images). TUVF also enables direct texture transfer through dense shape correspondence. Such property is also significant to the field because post-processing texture transfer often requires user-provided correspondence (which is extremely expensive) or ad hoc assumptions about the input geometry.

We summarize the differences in Table 10 below to clarify the distinctions between TUVF and recent works. To assess the realism of the generated textures compared to natural images, we follow Texturify's setup and evaluate them on the CompCars dataset, which contains real car photos from various angles. It is important to note that this is not an apples-to-apples comparison for baselines that are not generalizable or require 3D supervision. This comparison doesn't aim to claim superior texture quality from our method. Instead, it highlights the differing approaches of these works and TUVF, given their distinct setups. The results also underscore the challenge of learning realistic texture representations due to the absence of large-scale realistic texture-shape datasets.

Table 10: **Comparison to recent state-of-the-arts on texture synthesis.** G denotes *Generalizable*, indicating whether a method is encoder-based or optimization-based. U denotes *Unsupervised*, indicating whether the method is free from 3D paired data supervision. C denotes *Correspondence*, indicating whether the method provides dense correspondence for texture transfer. The inference time is measured in minutes.

| Method | G | U | C | FID ↓ | KID ↓ | Inference Time↓ |
|---|---|---|---|---|---|---|
| AUV-Net (Chen et al., 2022) | ✓ | | ✓ | N/A | N/A | N/A |
| Texturify (Siddiqui et al., 2022) | ✓ | ✓ | | 59.55 | 4.97 | 0.01 |
| TEXTure (Richardson et al., 2023) | | | ✓ | 181.67 | 14.09 | 5 |
| Point-UV Diffusion (Yu et al., 2023) | ✓ | | | 101.31 | 8.61 | 3.5 |
| Text2Tex (Chen et al., 2023) | | | ✓ | 46.91 | 4.35 | 15 |
| TUVF | ✓ | ✓ | ✓ | 41.79 | 2.95 | 0.01 |

## T  LIMITATIONS

**Geometry.**  Our work has some limitations inherited from Cheng et al. (2021) since our Canonical Surface Auto-encoder follows similar principles. Specifically, encoding the shape information of a point cloud in a global vector may cause fine details, such as corners and edges, to be blurred or holes to disappear after reconstruction. Similar to Cheng et al. (2021), we also observed that the correspondences predicted near holes or the boundaries between parts might be incorrect, possibly due to the sparsity nature of point clouds and the limitations of the Chamfer distance. Future research should address these limitations.

**Characteristic Seams.**  Seams are barely noticeable in our results. There are three reasons. Firstly, unlike prior works (Chen et al., 2022), we avoid cutting the shape into pieces and instead use a unified UV across all parts, resulting in a seamless appearance without any distinct boundaries. Secondly, our UV mapper employs a non-linear mapping function trained with Chamfer loss, seamlessly connecting the UV coordinates without explicit stitching lines. Thirdly, unlike prior works that directly regress RGB values using UV features or RGB information alone, our $MLP_F$ also takes the local coordinate as an additional input, representing a local radiance field that effectively reduces the seams. However,

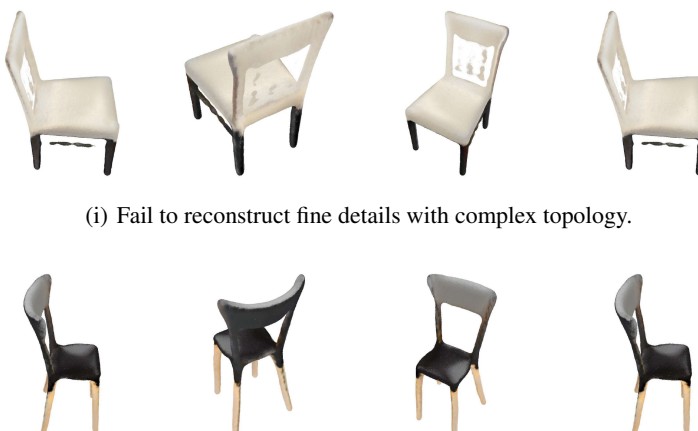

(i) Fail to reconstruct fine details with complex topology.

(ii) Incorrect correspondences near part boundaries.

Figure 30: **Visualization of Failure Cases.**

these design choices do not completely solve the seam issue. As illustrated in Figure 23, unsmooth correspondence can still result in visible seams.

