# OpenReview forum: "TUVF: Learning Generalizable Texture UV Radiance Fields"
_ICLR.cc/2024/Conference — ICLR 2024 poster_

### Official Review · Reviewer_K7dY · 2023-10-25

**Soundness:** 3 good
**Presentation:** 3 good
**Contribution:** 3 good
**Rating:** 8
**Confidence:** 3

**Summary:**

This work studies generating high-fidelity textures of 3D  shapes.

It introduces TUVF that generates textures in a learnable UV sphere space, which allows the texture to be disentangled from the underlying shape and transferable to other shapes from the same category.

It uses a sampled texture code that represents a particular appearance style adaptable to different shapes and generates the texture in a canonical UV sphere space. It learns a canonical surface auto-encoder that maps any point on a canonical UV sphere to a point and normal on an object’s surface, which is transformed to indicator function values using the Poisson Surface Reconstruction algorithm and further transformed to density. This step is learned by Chamfer Distance on the surface points and the L2 losses on the indicator grid. Finally, the texture is learned by neural rendering with a patch-based discriminator.

The correspondence between the UV space and the 3D shape is automatically established during training.

**Strengths:**

TUVF achieves much more realistic, high-fidelity, and diverse 3D consistent textures compared to previous approaches.

It achieves state-of-the-art results in both the experiments on synthetic and real-world object datasets.

It improves texture control and editing.

The use of the Poisson Surface Reconstruction algorithm is very interesting and contributes insights for the community.

**Weaknesses:**

There still exist distortions in texture.

The contribution of texture generation is limited due to recent text-driven texture synthesis work, such as Text2Tex and TEXTure: Text-Guided Texturing of 3D Shapes.

**Questions:**

My main concern is about the contribution. The contribution of this work versus the work of text-driven texture synthesis should be further clarified.

---

> ### Author Response · Authors · 2023-11-20
> **Author Response to Reviewer K7dY**
>
> **Q**: There still exist distortions in texture.
>
> **A**: While distortions are a common issue in UV-based methods, our approach has notably improved in reducing seams and distortions.
>
> The key factors contributing to this reduction include:
>
> * **Unified UV Mapping**: Unlike previous methods that cut shapes into pieces, we ensure that the UV mapping is shared across all parts of the shape. This results in a seamless appearance without distinct boundaries.
>
> * **Non-linear Mapping**: Our UV mapper employs a non-linear mapping function trained using Chamfer loss. This approach implicitly connects UV coordinates without the need for explicit stitching lines.
>
> * **Local Radiance Field**: Our MLP$_{F}$ incorporates local coordinates as an additional input, representing a local radiance field. This effectively minimizes seams in the generated textures.
>
> It's important to note that while these design choices significantly reduce seam and distortion issues, they may not entirely eliminate them. In the future, we could improve by using better data-driven techniques (e.g., diffusion prior) or more advanced neural rendering methods (e.g., ray transformers), as we mentioned in Section 5 of the main paper.
>
> ---
>
> **Q**: The contribution of texture generation is limited due to recent text-driven texture synthesis work, such as Text2Tex and TEXTure: Text-Guided Texturing of 3D Shapes.
>
> **A**: Please refer to the [General Response (B)](https://openreview.net/forum?id=dN4vpVTvWX&noteId=zW1K8ECNIl) for justification. We provide quantitative and qualitative evaluations and discussions highlighting the differences between recent text-driven texture synthesis works and TUVF. Our analysis concludes that these are distinct lines of work within TUVF (links: [AUV-Net](https://openreview.net/forum?id=dN4vpVTvWX&noteId=4Xylia94Au), [TEXTure](https://openreview.net/forum?id=dN4vpVTvWX&noteId=4Xylia94Au), [Point-UV Diffusion](https://openreview.net/forum?id=dN4vpVTvWX&noteId=uBGrwFvF1o), [Text2Tex](https://openreview.net/forum?id=dN4vpVTvWX&noteId=uBGrwFvF1o)).

---

> > ### Comment · Reviewer_K7dY · 2023-11-22
> >
> > I raise my rating. Thanks to the authors for the extra details in the rebuttal, particularly the comparison of Text2Tex and TEXTure in General Response (B). It would be helpful to include these and the reply of factors contributing to reducing seams and distortions in the camera-ready paper/supplement.

---

> > > ### Author Response · Authors · 2023-11-22
> > >
> > > Dear Reviewer K7dY,
> > >
> > > Thank you for your valuable feedback. We appreciate your time and effort in reviewing our modifications and responses. And we are glad that the questions have been resolved! **We will include all experiments and discussions in our revised paper/supplement.**

---

### Official Review · Reviewer_VxE5 · 2023-10-31

**Soundness:** 2 fair
**Presentation:** 3 good
**Contribution:** 2 fair
**Rating:** 6
**Confidence:** 4

**Summary:**

This paper proposes a method to learn a category-level latent space of both canonical surface and UV texture fields, from a 3D shape dataset and an unpaired collection of 2D images depicting this category of shapes. There are mostly three model components: (1) a canonical surface autoencoder that encodes the ground-truth shape into a latent code, which then gets decoded into points on a sphere, (2) a generative model that produces texture features conditioned on the style code, and (3) a differentiable radiance field rendering module that renders the generated texture onto the autoencoded shape, producing an image that can be compared against real images of the same shape category.

For the canonical surface autoencoder, Chamfer distance is used in this autoencoding task, and the latent code learned should ideally "instruct" us where on the canonical sphere each surface point falls onto, therefore providing a common space to anchor all shapes within the class (i.e., dense correspondence). The authors opted to turn point clouds into density volumes with differentiable Poisson surface reconstruction.

For the texture feature generator, the authors adopted StyleGAN-like style injection and made sure there's no interaction between neighboring pixels since vicinity in the UV space is often not physically meaning. RGB colors are not explicitly decoded from these features until the final rendering.

For the differentiable rendering module, the authors chose to adopt volume rendering even if the native shape representation is mesh from ShapeNet. Because of that, the authors discussed how to efficiently sample rays (rendering only near the object surface), convert point clouds into volume density (with differentiable Poisson surface reconstruction), and define radiance fields for points (via interpolating nearest surface points).

The authors show reasonable qualitative results where they can transfer texture to another shape in the category, make 3D-consistent texture edits, and find dense correspondence among shapes in the same class.

**Strengths:**

The paper does a good job presenting what is done with helpful visuals and is easy to follow.

It also tackles an interesting problem where one needs to learn a canonical space for a category shapes and simultaneously put textures onto the shapes, without paired 3D-2D data. By leveraging autoencoding and adversarial learning, the model learns meaningful patterns/correlations without explicit, direct supervision.

Dense correspondence emerging from autoencoding is also interesting and makes sense.

**Weaknesses:**

I have two significant concerns that need addressing before I can consider raising my ratings.

While I understand how canonical surface autoencoding eventually leads to a mapping between a given shape and the canonical sphere, for rigid shapes like cars and airplanes, one can project the shape onto an enclosing sphere, e.g., via raycasting from the sphere to the shape, to obtain similar mappings -- "similar" as in all cars' front bumpers mostly map to the same location on the canonical sphere. If this simple approach produces similar results, the whole canonical surface autoencoding part becomes invalid.

I understand the authors also show mappings learned for non-rigid objects like humans and animals, but these are all parametric models (SMPL or SMAL or whatever), so the canonical space for them is by definition well established and doesn't benefit from this work.

The second major concern is the whole deal of converting meshes or point clouds into density volumes. Volumetric approaches like NeRF are cool but I don't think everything needs to or should be volumetric. If we already have meshes, why throw away the face information, go into the point cloud regime, and then make the points volumetric? Each of these steps is lossy and complicates the method in an unnecessary way in my opinion. A concrete, much simpler alternative is retaining the face information for the points on the canonical sphere and use the same faces in the decoded shape. Then, we don't need differentiable surface reconstruction to return to the mesh domain in a suboptimal manner. What confuses me further is the adoption of volume rendering. With this alternative I proposed, one simply renders the mesh, which will be much more computationally efficient.

If these two concerns/confusions don't get cleared up, I view this paper as over-engineering a problem that could be tackled in a cleaner and simpler way, possibly also compromising the final quality given the extra lossy steps taken.

**Questions:**

As mentioned in "Weaknesses," can we produce a mapping without learning by just raycasting from an enclosing sphere or something similar? I can see self-occlusion might be a blocker, but the learned mapping is not perfect either; this simpler alternative seems to deserve a try.

Why do we go through the complicated pipeline to make the shape volumetric? This looks like an even more severe issue than the first question. I hope we have a good justification (or I'm misunderstanding the paper); our community would hate to see "volumetric == cool" as the motivation.

---

> ### Author Response · Authors · 2023-11-20
> **Author Response to Reviewer VxE5**
>
> **Q**: For rigid shapes like cars and airplanes, one can project the shape onto an enclosing sphere to obtain similar mappings -- "similar" as in all cars' front bumpers mostly map to the same location on the canonical sphere. If this simple approach produces similar results, the whole canonical surface autoencoding part becomes invalid.
>
> **A**: As per the reviewer's suggestion, we develope a new baseline method to establish a correspondence mapping by casting rays from a sphere to shapes. We test the baseline approach on the KeypointNet dataset and evaluate its effectiveness by conducting experiments on the semantic keypoint transfer task. The table below showcases the qualitative results we obtained on Cars. Our findings suggest that this method can generate reasonable results for some samples, but it is prone to errors when it comes to local areas due to sensitivity to scale and offset. Moreover, we have observed noticeable seams while performing texture transfer with synthetic textures, which we show an example [here](https://drive.google.com/file/d/10PhutLoShKmxcoO-Ck-RI0-YyLHdWroq/view?usp=drive_link).
>
> | ShapeNet Car (Image) | ShapeNet Car (Video) |
> | -------- | -------- |
> |[results](https://drive.google.com/file/d/1xyVqqlm2xI_FSOd1xhwlp1LfSx-M_uUW/view?usp=drive_link)|[results](https://tuvf4iclr.github.io/carkpt)|
>
>
> To further quantitatively evaluate the method, we measure the correspondence accuracy at different error thresholds, and the results are presented below. Additionally, we experimented with various radius parameters to improve the results. The findings indicate that the radius length significantly influences the correspondence accuracy. This is because only sparse points are cast to the surface when the sphere is large. And if the sphere is small, seams and occlusion will occur more frequently. Despite the variations in radius length, the Canonical Point Auto-encoder outperformed all other variants.
>
> | ShapeNet Airplane | ShapeNet Car | ShapeNet Chair |
> | -------- | -------- | -------- |
> | [results](https://drive.google.com/file/d/1BMD5FMIgmX6CQy0oAJt_6AalQDrzprpQ/view?usp=drive_link)     | [results](https://drive.google.com/file/d/1rc6oBhffNd-K_Nk7P7W6-p5xke8rouiX/view?usp=drive_link)     | [results](https://drive.google.com/file/d/1RBL5jQ64FJJdEHWekfG7ePWU-JuSfMhu/view?usp=drive_link)     |
>
> ---
> **Q**: I understand the authors also show mappings learned for non-rigid objects like humans and animals, but these are all parametric models (SMPL or SMAL or whatever), so the canonical space for them is by definition well established and doesn't benefit from this work.
>
> **A**: As mentioned in our main paper, we have also evaluated generic man-made objects, including cars and chairs. Since these are topology-varying objects, it is not easy to fit them with parameterized primitives. As a result, these objects lack a clear, well-defined canonical space. Therefore, our work is particularly useful for these objects.
>
> ---
> **Q**: Volumetric approaches like NeRF are cool but I don't think everything needs to or should be volumetric. If we already have meshes, why throw away the face information, go into the point cloud regime, and then make the points volumetric? Each of these steps is lossy and complicates the method in an unnecessary way in my opinion.
>
> **A**: We chose to use NeRF not because it's trendy but because it's the most intuitive option available. In [General Response (A)](https://openreview.net/forum?id=dN4vpVTvWX&noteId=HLOybPoVgu), we conducted a series of studies exploring alternative approaches with mesh representation. We found that without NeRF, it's not straightforward to maintain correspondence on mesh faces while ensuring good geometry. Therefore, using a continuous representation is a logical and intuitive choice and the best option available. We want to re-emphasize that no steps in our method are unnecessary. We use a point cloud regime because it can better represent the topology-varying geometry and correspondence compared to a mesh. Additionally, we make the points volumetric to maintain surface correspondence for rendering.
>
>
> ---
> **Q**: A concrete, much simpler alternative is retaining the face information for the points on the canonical sphere and use the same faces in the decoded shape. Then, we don't need differentiable surface reconstruction to return to the mesh domain in a suboptimal manner.
>
> **A**: We follow the reviewer's suggestion and develop a new mesh-based baseline that uses the face information from the canonical sphere to form the reconstructed shape. However, we find that such a design is not feasible. The Canonical Point Auto-encoder allows complex transformations like folding or cutting. Therefore, using the same faces as the points on the canonical sphere would lead to incorrect meshes. Please refer to [General Response (A)](https://openreview.net/forum?id=dN4vpVTvWX&noteId=HLOybPoVgu) for details and results.

---

> > ### Comment · Reviewer_VxE5 · 2023-11-22
> > **Raising my rating**
> >
> > I thank the authors for the additional experiments and helpful visualization to prove their points. Although I'm still slightly concerned about over-engineering, I'm convinced that this paper proposes a valid approach to a complex problem. Therefore, I'm willing to raise my rating provided that the authors will include all of these new experiments in their final manuscript.

---

> > > ### Author Response · Authors · 2023-11-22
> > >
> > > Dear Reviewer VxE5,
> > >
> > > Thank you for taking the time to provide us with your valuable feedback. We appreciate your effort in reviewing our paper and responses, and we are glad that your concerns have been resolved! **We will include all experiments and discussions in our final manuscript.**

---

> ### Author Response · Authors · 2023-11-22
>
> Dear Reviewer VxE5,
>
> Thank you once again for the detailed feedback.
> We are approaching the end of the author-reviewer discussion period. However, there are no responses yet to our rebuttal.
>
> Please feel free to request any additional information or clarification that may be needed. We hope to deliver all the information in time before the deadline.
>
> Thank you!

---

### Official Review · Reviewer_KY39 · 2023-11-01

**Soundness:** 3 good
**Presentation:** 2 fair
**Contribution:** 2 fair
**Rating:** 6
**Confidence:** 5

**Summary:**

This paper proposes a textured shape generation method based on learned UV mapper and neural texture features.
The training is supervised by images rendered on 2D space.
After training the method can support random textured generation for a given shape, texture editing, and texture transfer.
Though the proposed pipeline is supported with a large mount of qualitative and quantitative results.
I find is that the paper's technical contribution is little and the results shown in the paper are OK but not exciting(not comparable to recent diffusion based method).
Most aspects have already been explored by previous papers(some important references are missing).
In addition, the intuition of using neural radiance field is really unclear.
I do not feel using neural radiance field can bring any benefit since every property before the rendering is on the surface.
Thus, I am leaning towards rejection but also listen suggestions from other reviewers.

**Strengths:**

The paper is overall clear with a good structure.
Readers can follow the text easily.
The paper shows a lot of results and comparisons, which makes the pipeline more convincing.
Details of the network architecture are given in the supp, making reproduction easier.

**Weaknesses:**

The biggest issue is that the paper is not novel.
Most part in the paper has been explored in previous papers.
Though well combined, it only produces OK results instead of exciting results.
For example, recent PointUVDiffusion (Texture Generation on 3D Meshes with Point-UV Diffusion) can generate very realistic textures for a 3D shape.
Though this paper can support more things like texture transfer via its UV mapper.
But actually, correspondence between shapes can also be obtained by postprocessing.
So I am wondering how the method compares to PointUVDiffusion.

Another issue is the intuition of using NeRF.
Though NeRF is hot and can reconstruct 3D scenes very well and provide vivid results, for this given task, I strongly feel that NeRF is not necessary because everything before the rendering process is defined on the surface.
Why not use surface-based rendering?
Is there any benefit to using NeRF (for example, view-dependent texture)? I do not see such results.
For example, AUV-Net learns an aligned UV space, which is pretty similar to the proposed pipeline, though it does not use a neural radiance field.
So I think comparing it to AUV-Net should be necessary to prove the advantage of NeRF.

A formulation of the rendering process would be better in Sec. 3.2 or Sec3.3.

Reference:
SINE: Semantic-driven Image-based NeRF Editing with Prior-guided Editing Field

**Questions:**

See the weakness section.

---

> ### Author Response · Authors · 2023-11-20
> **Author Response to Reviewer KY39 (1/2)**
>
> **Q**: The biggest issue is that the paper is not novel. Most part in the paper has been explored in previous papers.
>
> **A**: We respectfully disagree with the reviewer. This comment can be applied to most deep-learning papers. Most new technologies proposed in the past 10 years in deep learning are based on existing tools used in previous papers. Thus, it is important to judge a work based on what new functionality has been achieved and how intellectually the techniques are used:
>
> -- **New Functionality**: TUVF is the first unsupervised work that enables generalizable texture synthesis with 3D dense correspondence at the same time. It not only achieves state-of-the-art texture synthesis results but also enables controllable synthesis by swapping texture while fixing the structure or the other way around, as well as transferring texture editing.
>
> -- **Technical Novelty**: TUVF is the first work that performs joint end-to-end Canonical Point Auto-encoder learning and surface learning. To perform rendering from the surface is non-trivial. Previous volumetric rendering in NeRF usually requires sampling in dense volume space, and our approach allows sampling near the surface. This not only largely improves the rendering efficiency but also leads to better texture. This results in a continuous field with dense and smooth correspondence on the surface, which is a representation never existed before. It also goes beyond texture synthesis and offers a new potential direction for efficient neural rendering.
>
> ---
>
> **Q**:  it only produces OK results instead of exciting results. For example, recent PointUVDiffusion (Texture Generation on 3D Meshes with Point-UV Diffusion) can generate very realistic textures for a 3D shape.
>
> **A**: We respectfully disagree with the reviewer's opinion that our results were described as "OK but not exciting." We believe that this statement is subjective and lacks specificity. For example, if we present texture synthesis results from the [Point-UV Diffusion sample](https://drive.google.com/file/d/14iMgNXLoXocWJXY8F9JkEHJXCE4I-6n6/view) and the [TUVF sample](https://tuvf4iclr.github.io/samples.html), which one can be viewed as performing better? Systematic quantitative and qualitative analysis is important for comparisons.
>
> Additionally, it would be unfair to compare TUVF with supervised methods that were trained on synthetic setups, such as Point UV Diffusion. In our [General Response (B)](https://openreview.net/forum?id=dN4vpVTvWX&noteId=zW1K8ECNIl), we have provided both quantitative and qualitative results. We also provide discussions that highlight the fundamental differences between TUVF and these recent approaches (links: [AUV-Net](https://openreview.net/forum?id=dN4vpVTvWX&noteId=4Xylia94Au), [TEXTure](https://openreview.net/forum?id=dN4vpVTvWX&noteId=4Xylia94Au), [Point-UV Diffusion](https://openreview.net/forum?id=dN4vpVTvWX&noteId=uBGrwFvF1o), [Text2Tex](https://openreview.net/forum?id=dN4vpVTvWX&noteId=uBGrwFvF1o)).
>
> ---
>
> **Q**: But actually, correspondence between shapes can also be obtained by postprocessing.
>
> **A**: We appreciate the reviewer's perspective but would like to emphasize the significance of learning dense correspondence in the context of 3D shape texture transfer. Learning dense correspondence for 3D shapes is a long-standing and challenging problem in the field. 3D shapes can be highly complex, with intricate details and variations. Notably, no heuristic-based tools can automatically and reliably establish dense correspondence between shapes without significant manual intervention. Meanwhile, manually assigning dense correspondence between such shapes can be extremely laborious and prone to error. This is unlike finding correspondence in 2D space, which can be achieved more conveniently by running DINOv2 given input images. Therefore, advancing our understanding of 3D dense correspondence remains a crucial and challenging goal.
>
> ---
>
> **Q**: The intuition of using NeRF. I strongly feel that NeRF is not necessary. Is there any benefit to using NeRF?
>
> **A**: Please see the [General Response (A)](https://openreview.net/forum?id=dN4vpVTvWX&noteId=HLOybPoVgu) for clarification. We conduct a series of studies exploring alternative approaches if not using NeRF. Moreover, we explain the motivation and intuition behind using NeRF. We find that without the help of NeRF, there is no straightforward way to maintain correspondence on the faces while ensuring good geometry. We chose to use NeRF not because it is trendy but because it is the most intuitive option available.

---

> ### Author Response · Authors · 2023-11-20
> **Author Response to Reviewer KY39 (2/2)**
>
> **Q:** Why not use surface-based rendering?
>
> **A**: We want to clarify that surface rendering techniques are already included as a crucial component of TUVF, which is discussed in detail in our paper and one of the **technical novelty** in this paper. Specifically, our approach involves a combination of surface and volume rendering methods. We render the color of a ray only on points near the object's surface, effectively leveraging the surface properties of the object. This approach results in superior rendering quality and efficiency. We kindly refer the reviewer to Section 3.3 of our main paper for more details.
>
> > **Efficient Ray Sampling.** Surface rendering is known for its speed, while volume rendering is known for its better visual quality (Oechsleetal.,2021). Similar to (Oechsleetal.,2021; Yarivetal.,2021; Wangetal.,2021), we take advantage of both to speed up rendering while preserving the visual quality, i.e., we only render the color of a ray on points near the object's surface.
>
> ---
>
> **Q**: Comparing TUVF to AUV-Net should be necessary to prove the advantage of NeRF.
>
> **A**: We appreciate the reviewer's interest in comparing our approach to AUV-Net to showcase the advantages of NeRF. However, it's important to note that AUV-Net is, again, a supervised approach with a synthetic setup. Another critical difference is their use of preprocessed point colors as supervisory signals. As a result, AUV-Net does not involve "rendering" during training. Therefore, direct comparisons between TUVF and AUV-Net may not effectively demonstrate the unique advantages of NeRF. NeRF's continuous property is one of the specific advantages that make it ideal for TUVF. Furthermore, NeRF's strength lies in its ability to synthesize high-quality images through rendering. Please refer to the [General Response (B)](https://openreview.net/forum?id=dN4vpVTvWX&noteId=zW1K8ECNIl) and [here](https://openreview.net/forum?id=dN4vpVTvWX&noteId=4Xylia94Au) for further discussion regarding AUV-Net.
>
> ---
>
> **Q**: A formulation of the rendering process would be better in Sec. 3.2 or Sec3.3.
>
> **A**: We have included the rendering process in Section 3.3 of our revised paper (colored in red). We use the quadrature rule to compute the color integral for each pixel.
>
> ---
>
> **Q**: Reference: SINE: Semantic-driven Image-based NeRF Editing with Prior-guided Editing Field
>
> **A**: We have included the recommended citation in our revised paper. However, we would like to clarify that the suggested paper mainly focuses on NeRF editing, which is less relevant to TUVF. Our primary objective is texture synthesis, while texture editing is a secondary outcome. Moreover, TUVF can directly transfer texture editing between shapes, which SINE paper cannot. We sincerely hope that this missing reference will not serve as grounds for rejection.

---

> ### Comment · Reviewer_KY39 · 2023-11-21
> **response**
>
> Yes. I knew that you used the efficient ray sampling strategy.
> However, the key problem is whether NeRF can bring better results in this task, for example, view-dependent appearance.
> Unfortunately, the paper does not contain such view-dependent results.
> Then why bother learning a view-dependent appearance function?
> Directly learning a color field is enough to produce the results in the paper.
> That is why I think it is not necessary.
> In addition, qualitative results with clear textures in AUV-NET are better than those in this paper though it requires GT 3D points‘ colors(TUVF requires points without colors).

---

> > ### Author Response · Authors · 2023-11-21
> > **Author Response to Reviewer KY39**
> >
> > Using 3D ground truths or not actually makes the key difference, and distinguishes our work from previous work. If we are able to do it without 3D ground truths, we can harvest large-scale data from real images on the web, instead of having an expert to make the simulated model for us each time. This is also what the learning community (ICLR conference) should care about most, about generalization and scaling, about how to use learning to achieve this, instead of getting very good results without caring how to achieve that.
> >
> > From the perspective of learning without 3D ground truths, how do we obtain supervision from images? The only way is through neural rendering; there is **NO OTHER WAY**. Our studies show NeRF provides a very good tool for obtaining supervision. NeRF is not just about view-dependent appearance (which is not mandatory in our network; see Section 3.3), it is about how you obtain supervision from 2D in any angles presented in the real world to learn 3D representation. We have provided detailed comparisons to other alternatives in general response. If the reviewer can take a closer look at the results of other approaches, it will be obvious that NeRF is the optimal solution and a necessity in solving our task, under the setting without 3D ground truths.

---

> > ### Author Response · Authors · 2023-11-22
> > **Response to our latest rebuttal**
> >
> > Dear Reviewer KY39,
> >
> > Thank you once again for providing us with detailed feedback. We are nearing the end of the author-reviewer discussion period, which is less than 24 hours away. However, we have not received a response to our latest rebuttal yet. We would like to know if our explanation was satisfactory for you.
> >
> > Please do not hesitate to ask for any additional information or clarification that you may require. We aim to provide all the necessary information before the deadline.
> >
> > Thank you.

---

> ### Comment · Reviewer_KY39 · 2023-11-22
>
> I agree that TUVF outperforms other unsupervised baselines.
> However, the answer still does not directly resolve my concerns about why using NeRF can get better results.
> Readers can only see the results and say OK your method gets better but miss intuition/reason.
> I really would like to see the authors replace the NeRF rendering module in TUVF with a simple color field(it's still neural rendering but not volume rendering) without sampling points near the surface and taking view direction as input. (that is to say, just predict the color on the surface.)
> Will there be difference on the quality side?
> If there are differences, then maybe we can say the sampling process can better depict texture details or something (hopefully I can raise my score).
> Otherwise, I really do not feel that is necessary.

---

> > ### Author Response · Authors · 2023-11-22
> >
> > We had ablated with different numbers of points sampled for rendering. Our findings showed that using only a single point, which is referred to as surface rendering, can lead to degraded results. This is because the color (UV feature) is obtained through KNN aggregation$^{*}$ (Sec 3.3 Point-based Radiance Field); using a single point may lead to instability and discontinuity at the boundary of two clusters. Therefore, in our paper, we opted to use multiple sampling points to obtain better results. The design of TUVF has a clear motivation that aligns well with our goals (as explained in the general response) and produces higher-quality results compared to alternatives.
> >
> > |CompCars | FID   | KID  |
> > |-------------------|-------|------|
> > | Surface Rendering | 58.60 | 4.86 |
> > | Ours (32 points)  | 41.54 | 3.01 |
> > | Ours (3 points)   | 41.79 | 2.95 |
> >
> > #### $^{*}$Based on Table 7 in Appendix H, in order to maintain the smoothness of the UV, it is necessary to learn the mapping from UV to the surface. This means that there is no direct definition of color (UV feature) on surface points. Therefore, when rendering the surface, KNN aggregation is still required to obtain the UV feature.

---

> > > ### Comment · Reviewer_KY39 · 2023-11-22
> > >
> > > With the additional ablation, I raised my score.

---

### Official Review · Reviewer_9F5H · 2023-11-01

**Soundness:** 3 good
**Presentation:** 4 excellent
**Contribution:** 3 good
**Rating:** 8
**Confidence:** 4

**Summary:**

This paper proposes a novel approach for generating novel texture given a 3D shape. The key idea of this paper is to learn correspondence between UV coordinates and 3D points and apply a generative model in the UV space. The correspondence associates the generated texture features with the actual 3D location on the surface. It enables differential volume rendering wrt. the texture features. The authors use an adversarial learning setup on respective renderings.
For the geometry, they used the 3D cars and chairs from the ShapeNet dataset and photoshapes and CompCars datasets are used as 2D GT. A qualitative and quantitative comparison to Texturify, EpiGraf and more baselines show a decent performance.

**Strengths:**

The paper is well written and easy to understand. The method sections constraints useful figures and clear structure.
The research problem is important since a general formulation of UV mapping is an open topic.
The experimental sections contain many insights and support claims, e.g.Table 4 the ablation on the texture mapping network.

**Weaknesses:**

Even though the method requires a GT shape as input, the rendered shapes appear to have over-smoothed regions, e.g. the mirrors of the cars.

It is unclear how well the learned UV correspondence preserves surface areas in the UV space.

**Questions:**

Since the proposed method uses a shape encoder decoder part, I’m wondering if this could be directly used to build a generative model for both shapes and textures.

---

> ### Author Response · Authors · 2023-11-20
> **Author Response to Reviewer 9F5H**
>
> **Q**: The rendered shapes appear to have over-smoothed regions. It is unclear how well the learned UV correspondence preserves surface areas in the UV space.
>
> **A**: We conduct experiments to evaluate our shape reconstruction method using the standard evaluation on ShapeNet, following previous studies. The table below shows the CD (Chamfer Distance) and EMD (Earth Mover's Distance) scores, comparing our method with several prior works. We also include the lower bound of the reconstruction errors in the "Oracle" column. Our results indicate that the geometry is similar to the oracle and comparable with previous works. When measured by EMD, our method consistently outperforms other methods, suggesting that our reconstructed point clouds have more uniformly distributed points on the surface. It is worth noting that EMD is considered a better metric to measure a shape's visual quality as it requires the outputs to have the same density as the ground-truth shapes. However, there is still room for improvement, as discussed in Appendix R. We will leave it as future work.
>
> | Dataset  | Metric | l-GAN (CD) [10] | l-GAN (EMD) [10] | AtlasNet (Sphere) [11] | AtlasNet (Patchs) [11] | PointFlow [12] |   ShapeGF [13] | DiffusionPM [14] |      Ours | _Oracle_ |
> |----------|--------|-----------:|------------:|------------------:|------------------:|----------:|----------:|------------:|----------:|---------:|
> | Airplane | CD     |      1.020 |       1.196 |             1.002 |             0.969 |     1.208 | **0.960** |       0.997 |     0.975 |  _0.837_ |
> |          | EMD    |      4.089 |       2.577 |             2.672 |             2.612 |     2.757 |     2.562 |       2.227 | **2.102** |  _2.062_ |
> | Chair    | CD     |      9.279 |       11.21 |             6.564 |             6.693 |    10.120 | **5.599** |       7.305 |     6.544 |  _3.201_ |
> |          | EMD    |      8.235 |       6.053 |             5.790 |             5.509 |     6.434 |     4.917 |       4.509 | **4.185** |  _3.297_ |
> | Car      | CD     |      5.802 |       6.486 |             5.392 |             5.441 |     6.531 |     5.328 |       5.749 | **5.178** |  _3.904_ |
> |          | EMD    |      5.790 |       4.780 |             4.587 |             4.570 |     5.138 |     4.409 |       4.141 | **3.555** |  _3.251_ |
>
> ---
>
> **Q**: Build a generative model for both shapes and textures.
>
> **A**: We appreciate the suggestion. However, it's important to note that in TUVF, we use a disentangled representation that enables us to sample texture latents from its posterior distributions without any interference from the shape. This disentangled representation allows more controllable synthesis. We can either swap the geometry or swap the texture without interfering with the other, which is hard to achieve with a unified representation. One possible solution is to combine TUVF with a pre-trained shape generative model like CanonicalVAE [15]. CanonicalVAE is a recent work that expands canonical point auto-encoder into a generative model with dense correspondence. This could be a promising direction for our future research.
>
>
>
> ---
> *[10] Panos Achlioptas and Olga Diamanti and Ioannis Mitliagkas and Leonidas Guibas. Learning Representations and Generative Models for 3D Point Clouds. International Conference on Machine Learning (ICML), 2018.*
>
> *[11] Thibault Groueix and Matthew Fisher and Vladimir G. Kim and Bryan C. Russell and Mathieu Aubry. AtlasNet: A Papier-Mâché Approach to Learning 3D Surface Generation. IEEE Conference on Computer Vision and Pattern Recognition (CVPR), 2018.*
>
> *[12] Guandao Yang and Xun Huang and Zekun Hao and Ming-Yu Liu and Serge Belongie and Bharath Hariharan. PointFlow: 3D Point Cloud Generation with Continuous Normalizing Flows. IEEE International Conference on Computer Vision (ICCV), 2019.*
>
> *[13] Ruojin Cai and Guandao Yang and Hadar Averbuch-Elor and Zekun Hao and Serge Belongie and Noah Snavely and Bharath Hariharan. Learning Gradient Fields for Shape Generation. European Conference on Computer Vision (ECCV), 2020.*
>
> *[14] Shitong Luo and Wei Hu. Diffusion Probabilistic Models for 3D Point Cloud Generation. IEEE Conference on Computer Vision and Pattern Recognition (CVPR), 2021.*
>
> *[15] An-Chieh Cheng, Xueting Li, Sifei Liu, Min Sun, Ming-Hsuan Yang. Autoregressive 3D Shape Generation via Canonical Mapping. European Conference on Computer Vision (ECCV), 2022.*

---

> ### Author Response · Authors · 2023-11-22
> **Response to our Rebuttal**
>
> Dear Reviewer 9F5H,
>
> Thank you once again for the detailed feedback. We are approaching the end of the author-reviewer discussion period (less than 24 hours). However, there are no responses yet to our rebuttal.
>
> Please feel free to request any additional information or clarification that may be needed. We hope to deliver all the information in time before the deadline.
>
> Thank you!

---

### Author Response · Authors · 2023-11-20
**General Response (1/5)**

We appreciate the reviewers for recognizing the importance of the research problem that our paper addresses (Reviewer `9F5H`, `VxE5`) and for finding our approach to be interesting (Reviewer `K7dY`, `VxE5`), novel (Reviewer `9F5H`), and insightful (Reviewer `K7dY`, `9F5H`). In this response, we provide a general response to the reviewers' feedback, focusing on two aspects: the motivation behind using the neural radiance field and the contribution of our work compared to recent texture synthesis works.
## **(A) Motivation for using the neural radiance field** [`KY39`,`VxE5`]
Our use of NeRF representation is highly motivated and intuitive rather than just a trendy choice. It provides a necessary mechanism for learning surface representation from pixels without any paired (mesh, textured mesh) training data. To understand why we opted for NeRF, here we discuss other options using mesh representation and their limitations. We find that without the help of NeRF, **_there is no straightforward way to maintain correspondence on the faces while ensuring good geometry_**.

1. **Direct Mesh Deformation**
One way to build dense correspondence using mesh representation is through direct deformation. This approach involves deforming a template mesh with the offset of each vertex while maintaining the mesh's topology. We show the results below using direct mesh deformation targeting [this chair](https://drive.google.com/file/d/1DMrTh0grvnpPIYS3rGLd17wwLAu9FDio). The main limitation of this approach is that it struggles with non-genus-zero shapes and tends to produce non-smooth results, even with various shape regularizers. We tested the method with multiple shape regularizers, including $L_{edge}$, $L_{normal}$, and $L_{laplacian}$, with different balancing weights. We also included images of the results when no regularizer was used (Vanilla).
    | Method | Deformation (Image) | Deformation (Video) |
    | -------- | -------- | -------- |
    Vanilla (w/o reg)| [results](https://drive.google.com/file/d/1E6ifAkLNDMMoImCGlPnxh1B8KlySXpSo) | [results](https://drive.google.com/file/d/1DCwH-mSroJXn2EMl2YJFrVr0nC9-RN5m) |
    Moderate Regularized | [results](https://drive.google.com/file/d/13tSf08rXvtulSTlVgfwpXXbG_BFM-fDw) | [results](https://drive.google.com/file/d/1d0fFX-Ja3-XRUQA01mGlt9gczQCZi2Bs) |
    Over Regularized| [results](https://drive.google.com/file/d/1ptxNpxDVA4Nqw72K5WoSo8rb0czHkR_D) | [results](https://drive.google.com/file/d/1-tCmJTbLnFCA_7mNRk5U3m9b4MmCTQrz) |
2. **Canonical Point Auto-encoder + Sphere Faces**
Canonical Point Auto-encoder can handle non-genus-zero shapes by allowing complex transformations like folding, cutting, squeezing, and stretching. Therefore, using the same faces as the points on the canonical sphere would lead to incorrect meshes. We show the results [here](https://tuvf4iclr.github.io/spherefaces).
3. **Canonical Surface Autoencoder + Marching Cubes**
While this method maintains correspondence for points and can reconstruct shapes that are not genus-zero, the faces generated by marching cubes lack correspondence. Therefore, these faces cannot be used for our purpose.

These challenges led us to choose the neural radiance field over the mesh representation for TUVF. The neural radiance field offers a continuous representation that aligns naturally with our design goals. Given that our canonical point auto-encoder already provides continuity by mapping every point on the sphere to the shape's surface, using a continuous representation is a logical and intuitive choice. To further ensure the continuity, we introduce an implicit CIPS-UV generator to achieve high-resolution and high-fidelity output.

Moreover, TUVF is not a simple adoption of NeRF. We use a combination of surface and volume rendering, rendering the color of a ray only on points near the object's surface. This approach effectively capitalizes on the surface properties of the object and results in superior quality and efficiency, as elaborated in Section 3.3 of our main paper.

---

> ### Author Response · Authors · 2023-11-20
> **General Response  (2/5)**
>
> ## **(B) Contribution compared to recent works** [`KY39`,`K7dY`]
>
> Texture synthesis is a popular field, with several recent works dedicated to it. While they all aim to synthesize high-quality textures for given shapes, they each have different setups. Some prioritize per-object optimization, some lack correspondence capabilities and others demand 3D supervision. TUVF stands out by focusing on learning generalizable, disentangled, and realistic textures for untextured collections of 3D objects in an unsupervised manner (without paired training data for supervised learning). What sets TUVF apart is its reliance on easily accessible images and untextured shape datasets, eliminating the need for explicit 3D color supervision. This is crucial because, to our knowledge, **_no large-scale realistic textured 3D dataset is currently available_**. Even recent efforts like Objaverse [1] (e.g., [Objaverse car](https://objaverse.allenai.org/explore?menu%5BlvisCategories%5D=car_%28automobile%29&page=1), and [Objaverse chair](https://objaverse.allenai.org/explore?menu%5BlvisCategories%5D=chair&page=4)), while providing substantial 3D data, have limited inter-class sample numbers (in the hundreds), and most of the textures are not photorealistic (comparing to natural images). TUVF also enables direct texture transfer through dense shape correspondence. Such property is also significant to the field because post-processing texture transfer often requires user-provided correspondence (which is extremely expensive) or ad hoc assumptions about the input geometry.
>
> We summarize the differences in the table below to clarify the distinctions between TUVF and recent works. To assess the realism of the generated textures compared to natural images, we follow Texturify's setup and evaluate them on the CompCars dataset, which contains real car photos from various angles. It is important to note that this is not an apples-to-apples comparison for baselines that are not generalizable or require 3D supervision. **_This comparison doesn't aim to claim superior texture quality from our method. Instead, it highlights the differing approaches of these works and TUVF, given their distinct setups._** The results also underscore the challenge of learning realistic texture representations due to the absence of large-scale realistic texture-shape datasets.
>
> We provide a detailed discussion and some samples for each work below. We also provide a [video](https://tuvf4iclr.github.io/samples.html) with samples generated by TUVF for comparison to other methods.
>
> | Method | Venue | Generalizable (w/o per-shape optimization) | Unsupervised (w/o textured 3D shapes as input)| Correspondence | CompCars FID $\downarrow$ |  CompCars KID (×$10^{-2}$) $\downarrow$ | Inference Time (min/ object) $\downarrow$|
> | -------- | --------   | -------- | -------- | -------- | -------- | -------- | -------- |
> | AUV-Net [2]|CVPR'22|✓| |✓| N/A  | N/A |N/A|
> | Texturify [3]|ECCV'22|✓|✓| |59.55 | 4.97|<0.01|
> | TEXTure [4]|SIGGRAPH'23 | |✓| |181.67| 14.09|~5|
> | Point-UV Diffusion [5]|ICCV'23|✓| | |101.31| 8.61|~3.5|
> | Text2Tex [6]|ICCV'23||✓||46.91 | 4.35|~15|
> | [TUVF](https://tuvf4iclr.github.io/samples.html)||✓|✓|✓|**41.79** | **2.95**|**<0.01**|

---

> ### Author Response · Authors · 2023-11-20
> **General Response (3/5)**
>
> ---
>
> ### **AUV-Net**
> We have discussed AUV-Net in the related work section of our main paper. Unfortunately, the code for AUV-Net has not been made public. We had to rely on the results presented in the AUV-Net paper to compare our results with theirs. However, their experiments involved six datasets, four of which (Renderpeople [7], Triplegangers [8], Turbosquid Cars [9], and Turbosquid Animals) are not publicly available. The other two datasets (ShapeNet Cars & Chairs) are publicly available. Still, they have been re-processed into simple geometry by the author (simplifying the meshes to reduce geometric details and baking the geometric details into textures), making it difficult to compare our results to those presented in their paper directly. Most importantly, the need for 3D supervision in AUV-Net sets it apart from our approach. Specifically, AUV-Net use preprocessed point colors as supervisory signals. Therefore, AUV-Net does not involve the process of rendering during training.
>
> ---
>
> ### **TEXTure**
> TEXTure is also capable of texture transfer. However, for each instance to be transferred, their approach must augment the source mesh and generate a new dataset to fine-tune a depth-conditioned diffusion model. Therefore, the representation is not fully disentangled, and the process incurs significant computational costs. In contrast, TUVF builds explicit surface-to-surface mapping for direct texture transfer, ensuring the complete separation of texture from geometry and enabling on-the-fly transfers. We use their official implementation to generate various samples for a car object, experimenting with multiple prompts to achieve realistic results. Despite our best prompt engineering efforts, the resulting textures, while of good quality, still exhibit a somewhat synthetic appearance (e.g., [#2](https://drive.google.com/file/d/1CYkCu80iM_cKVCpPACGzWVY6dNBgQ-Im), [#4](https://drive.google.com/file/d/10yc_btD3m1FU0zmVWNK_zvsQFQEik4v1)). As a result, they yield a higher FID score when compared to natural images.
>
>
> | Prompt | Generated Samples (Image Grid) | Generated Samples (Video) |
> | -------- | -------- | -------- |
> | _A realistic car._  | [#1](https://drive.google.com/file/d/11hH9jXMYv1RUP0gjMtuvhrPxUUhaGKY_),[#2](https://drive.google.com/file/d/1b82uMZ8ef6sbBOzhJ-HTmuZ9s8oB1zIK),[#3](https://drive.google.com/file/d/1Ma4ojAorkMftssVDTAukQ6xZ_1Ie8OgM)| [#1](https://drive.google.com/file/d/1cDTKys4EQjGChs8Wix_LZuatzGsGXoja),[#2](https://drive.google.com/file/d/1GxPl9jkV8BJdrOrp9FXeo2ms-HeUt_xm),[#3](https://drive.google.com/file/d/1F8qYWmBt06SODyk9Yef-NMYr9V3IPM7G)|
> | _A realistic car, honda._  | [#1](https://drive.google.com/file/d/1KnlcacOJwpG7g40jUot7cESLSlRiZOkJ),[#2](https://drive.google.com/file/d/15QxwkzemunKb1fUM9h8RJAKZDZtW95WY),[#3](https://drive.google.com/file/d/1iu-y_MW8gRN8aagz147UoCY6jYvY-ix9)| [#1](https://drive.google.com/file/d/139Xrk8yVPbDGecaBbzaRFgB3QRshhod9),[#2](https://drive.google.com/file/d/1k8ivEq2sHlCARtXQf3LxioSJfbvNlApE),[#3](https://drive.google.com/file/d/1mUZ9xOH34FqyUd0r8tqJtFX9ut0pXnsZ)|
> | _A DLSR image of a real car, honda, realistic._  | [#1](https://drive.google.com/file/d/1QbNZW5r09fILsl53Hf3kGj_U9qC7uktD),[#2](https://drive.google.com/file/d/1sStgJDi86fgnBXM3GgVjVuh7RjWxVRXG),[#3](https://drive.google.com/file/d/1Qh2BU0MuWfrasU9-RudGSOtdDhHKidOd)| [#1](https://drive.google.com/file/d/1d1qr8TH5pIC7jTFCgrudEowttHRyVXZv),[#2](https://drive.google.com/file/d/1VgcNLP9_5rQZB4YabCqxEklujyHVg2I8),[#3](https://drive.google.com/file/d/1NewTMllSIIoyziSmi_3aUkcS_x5O56Ph)|
> | _A DLSR image of a real car, honda, realistic, high quality, high res._  | [#1](https://drive.google.com/file/d/1pFT9A-14Pa7uW-JtVAszCgvFnHj7Ncoi),[#2](https://drive.google.com/file/d/1CYkCu80iM_cKVCpPACGzWVY6dNBgQ-Im),[#3](https://drive.google.com/file/d/1KIAZ0yA1v7iaMB9TR3rD6qhfi92iUPp-),[#4](https://drive.google.com/file/d/10yc_btD3m1FU0zmVWNK_zvsQFQEik4v1)| [#1](https://drive.google.com/file/d/1JKBbz-ZFbrWbI9HthTE2f98HYIyseOFX),[#2](https://drive.google.com/file/d/1kiVMEH2JHwOFduSDVd_BQHIkiDCbKi6G),[#3](https://drive.google.com/file/d/1-B01RGTS-QXYas6gt9uqxcRLjmU9gp8X),[#4](https://drive.google.com/file/d/1L0n0_d9WzH18EQIhIf-1D1ueRpseq9KL)|

---

> ### Author Response · Authors · 2023-11-20
> **General Response (4/5)**
>
> ---
>
> ### **Point-UV Diffusion**
> Point-UV Diffusion is similar to AUV-Net, as it also requires 3D supervision in the form of textured shapes. We use their official implementation and the fine car checkpoint trained on ShapeNet. The rendering results are close to the ShapeNet ground truth. However, the method produces high FID scores compared to natural car images because ShapeNet textures are of low resolution, and the textures are often overly simplistic [1]. Additionally, although the method is trained in a generalizable manner, its inference time is high due to its coarse-to-fine design and long diffusion steps. Moreover, although the proposed method is based on UV, there is no dense correspondence between shapes, and therefore, it does not provide texture transfer ability.
>
> | Generated Samples (Image Grid) | Generated Samples (Video) |
> | -------- | -------- |
> |[#1](https://drive.google.com/file/d/1m8YHFTPFVZQliaFNOnzZQZ14e0DpXzaI),[#2](https://drive.google.com/file/d/1kcrQV5zYOO0WJQhFs2IzSrHk2pOSKlOU),[#3](https://drive.google.com/file/d/15d8ovL7omf8NFg4cKNFdEjsxj7CR23oW),[#4](https://drive.google.com/file/d/1cqYeQuMa0N-wgMJDMtGC02difzA5HtmN),[#5](https://drive.google.com/file/d/1HDhkbjIXVvHTz7yo_su7deNWvtS7Mnx3),[#6](https://drive.google.com/file/d/1zeskJeDakD2VkIRO5BbB5DUwTirqDtBb),[#7](https://drive.google.com/file/d/1Bewmr7b6MwOF40PVFK_vSDLZgndAVxvD),[#8](https://drive.google.com/file/d/1gLr_pJF1S-JAMuBcgle1CDqGDHH-sj4b),[#9](https://drive.google.com/file/d/1253cswhunIkFADkYVbG-00qJbOYg4Oby),[#10](https://drive.google.com/file/d/1LaVBJoYj2fg-CefjCdM-kDRHo3cBkdKd)| [#1](https://drive.google.com/file/d/14iMgNXLoXocWJXY8F9JkEHJXCE4I-6n6),[#2](https://drive.google.com/file/d/15pyZ3MzlnjTi1tgfS30uoZQ08tpo0O9I),[#3](https://drive.google.com/file/d/1fXRY6rFzlC79ddMm98xgpe5L_VufHL-_),[#4](https://drive.google.com/file/d/1Qd9ET0DjLI2KrCn-nyO9-33ePp-pJieA),[#5](https://drive.google.com/file/d/1Dpr6boHA52Wqt0iQDN3Win4-vgC-uaJ3),[#6](https://drive.google.com/file/d/1lkHjYlqZ5iUpXPBgQS38YEB-3q3xxu_9),[#7](https://drive.google.com/file/d/19iJSFsJXiufDwOTd_6bloxnj9CcnoGg4),[#8](https://drive.google.com/file/d/1OUKHbY1davzsPIFpO8eeqytvi1xXCanw),[#9](https://drive.google.com/file/d/1407BUOU5wWivaOrNCBFE1sCIJRNvQB1_),[#10](https://drive.google.com/file/d/1cUQiKGF-SMcNdaEN9tvwU69ro0RUOSNG)|
>
> ---
>
> ### **Text2Tex**
> Text2Tex employs per-scene optimization using a pre-trained text-to-image diffusion model, which comes with the drawback of being relatively slow, taking approximately 15 minutes to process a single object. While it can generate fairly realistic textures, it lacks correspondence capabilities, making it unsuitable for texture transfer tasks. We use their [official implementation](https://github.com/daveredrum/Text2Tex) with various prompts and seeds to provide samples below. We directly report the numbers from the CompCars evaluations in their paper.
>
> | Prompt | Generated Samples (Image Grid) | Generated Samples (Video) |
> | -------- | -------- | -------- |
> | _A realistic car._  | [#1](https://drive.google.com/file/d/1keqo90JSgKH0hotq3KHf5Ktzy3H70tR4),[#2](https://drive.google.com/file/d/1toR2tzQCamtDXpsW_gTetXjryQD3kcSk),[#3](https://drive.google.com/file/d/1fGkrRyPXBTxksAik0RUQcltQvBCtvUaN)| [#1](https://drive.google.com/file/d/193RN9zlFr8r2yocOfCmZ1YD5HivUD-pf),[#2](https://drive.google.com/file/d/1sUtyj9oLv5YJaPQX4A3pNSmlQzjAJPXM),[#3](https://drive.google.com/file/d/1IxdgjD1_ALe1fNSXnpBjaUYSlwud8DIE)|
> | _A realistic car, honda._  | [#1](https://drive.google.com/file/d/1pnLBJ8CPeKRdZS_bd_4iY1rZLoNwVHvr),[#2](https://drive.google.com/file/d/1avvyvtpwtFpYM2dMpstdd8Ujfvj2kcC3),[#3](https://drive.google.com/file/d/1elXI1i0vwt2kT5f2SC5DulmP1KnxTbqh)| [#1](https://drive.google.com/file/d/1s1cZ5XLA4L8lCWOL9XH7dwmSAYUYLfjm),[#2](https://drive.google.com/file/d/1LIy6qMM2B87I4xnORw9H558PYdJD71wR),[#3](https://drive.google.com/file/d/1x4tQJzGUI4VT5HjXuWdXyjW7_FOXWhOb)|
> | _A DLSR image of a real car, honda, realistic, high quality, high res._  | [#1](https://drive.google.com/file/d/1Vg1MzABEsSH9xUoH685EnL0Su2n40x3V),[#2](https://drive.google.com/file/d/17oVWZHYscYBCOrdfAbCcxGcFkvRRUZqw),[#3](https://drive.google.com/file/d/1UF2X8dhyHU1A6sKIfseoxd_8Lcl7h1FU)| [#1](https://drive.google.com/file/d/15vpWMn4G0dsv47m_OYXRBg4ASgLwB1hA),[#2](https://drive.google.com/file/d/1CnIdWSCaOxBPGpUg7eTKtvN1eOJRPq_l),[#3](https://drive.google.com/file/d/17JtUFNSRgMGtdCCj9FhbhaTQaxwbQWKY)|

---

> ### Author Response · Authors · 2023-11-20
> **General Response  (5/5)**
>
> ---
> **References**
>
> *[1] Matt Deitke and Dustin Schwenk and Jordi Salvador and Luca Weihs and Oscar Michel and Eli VanderBilt and Ludwig Schmidt and Kiana Ehsani and Aniruddha Kembhavi and Ali Farhadi. Objaverse: A Universe of Annotated 3D Objects. IEEE Conference on Computer Vision and Pattern Recognition (CVPR), 2023.*
>
> *[2] Zhiqin Chen and Kangxue Yin and Sanja Fidler. AUV-Net: Learning Aligned UV Maps for Texture Transfer and Synthesis. IEEE Conference on Computer Vision and Pattern Recognition (CVPR), 2022.*
>
> *[3] Yawar Siddiqui and Justus Thies and Fangchang Ma and Qi Shan and Matthias Nießner and Angela Dai. Texturify: Generating Textures on 3D Shape Surfaces. European Conference on Computer Vision (ECCV), 2022.*
>
> *[4] Elad Richardson and Gal Metzer and Yuval Alaluf and Raja Giryes and Daniel Cohen-Or. TEXTure: Text-Guided Texturing of 3D Shapes. SIGGRAPH, 2023.*
>
> *[5] Xin Yu and Peng Dai and Wenbo Li and Lan Ma and Zhengzhe Liu and Xiaojuan Qi. Texture Generation on 3D Meshes with Point-UV Diffusion. IEEE International Conference on Computer Vision (ICCV), 2023.*
>
> *[6] Dave Zhenyu Chen and Yawar Siddiqui and Hsin-Ying Lee and Sergey Tulyakov and Matthias Nießner. Text2Tex: Text-driven Texture Synthesis via Diffusion Models. IEEE International Conference on Computer Vision (ICCV), 2023.*
>
> *[7] RenderPeople. [https://renderpeople.com/](https://renderpeople.com/).*
>
> *[8] Triplegangers. [https://triplegangers.com/](https://triplegangers.com/).*
>
> *[9] Turbosquid. [https://www.turbosquid.com/](https://www.turbosquid.com/).*

---

> ### Comment · Reviewer_KY39 · 2023-11-21
> **TUVF also requires 3D points as input.**
>
> `Unsupervised (w/o textured 3D shapes as input)` column.
> Correct me if I am wrong, TUVF also requires 3D points without colors as input.
> The input difference between TUVF and AUV-NET is whether there are colors for points.
> But TUVF also requires multi-view images right?
> It's hard to say which requires more input.

---

> > ### Author Response · Authors · 2023-11-21
> > **Author Response to Reviewer KY39**
> >
> > All texture synthesis papers need shapes as inputs to output textures. The main difference between AUV-Net and TUVF is the need for 3D ground truth texture supervision. AUV-Net requires an expert to design the texture for every shape, providing a paired "shape+texture" mesh as ground truth. On the other hand, TUVF is fully unsupervised, without the need for ground truth texture in any form. TUVF does not require rendered images of "shape+texture" paired 3D objects. The supervision for TUVF comes from 2D image collections that are unrelated to the input shapes, such as real images on the web, like ImageNet.

---

> > > ### Comment · Reviewer_KY39 · 2023-11-22
> > >
> > > OK. This point is clearer to me now. TUVF does not require paired data since it only uses GAN loss.

---

### Meta-Review · Area_Chair_HjcA · 2023-12-07

**Metareview:**

This paper proposes an approach for generating novel category-level textures given a 3D shape of a category. The approach synthesizes category-level textures by training on a collection of single-view images of the category. The method learns correspondences between UV coordinates and 3D points and then applies a generative model to the UV space. There are three modules: a canonical surface autoencoder encoding the shape into a latent code, a generative model generating texture features based on the style code, and a differentiable radiance field renderer rendering the generated texture onto the autoencoded shape for synthesis. In addition to generating realistic category-specific textures, the method can also be used to generate, edit, and transfer textures.

Reviews recognize that UV mapping formulation is an important research problem. Technically, the paper addresses an interesting problem in which one needs to learn a canonical space for a category of shapes and simultaneously apply textures to those shapes without having paired 3D-2D data. Using autoencoding and adversarial learning, the model can learn meaningful patterns and correlations without explicit, direct supervision from 2D image collections that are unrelated to the input shapes, such as real images found on the internet. It is also generally praised that the experiments show more realistic, high-fidelity, and diverse 3D consistent textures in comparison to previous approaches. Based on both qualitative and quantitative comparisons, the proposed method outperforms baseline methods, such as Texturify and EpiGraf.

The initial reviews raised a number of concerns. It is unclear whether the proposed method is superior to recent diffusion-based methods. The rebuttal provides comparisons and demonstrates that the proposed method produces more realistic results. Secondly, reviews question how the proposed method compares to PointUVDiffusion and AUV-Net. In rebuttal, it is argued that these methods are supervised while the proposed method is unsupervised since it only uses unpaired 2D images. Reviewers also questioned its novelty as most parts have been explored in other papers. The rebuttal argues that the method is the first unsupervised method that is capable of generalizing texture synthesis with dense 3D correspondence. It also provides joint end-to-end Canonical Point Auto-encoder learning and surface learning. Last but not least, there is a question as to whether neural radiance fields are actually necessary. A surface-based rendering method is mentioned as an alternative. Rebuttal provides an ablation study that demonstrates that NeRF is superior to surface-based rendering. Moreover, it explains that the proposed method actually combines both surface and volume rendering techniques. An essential function of NeRF is to obtain supervision from 2D images in any angle to learn 3D representation while maintaining correspondences on the faces and ensuring that the geometry is correct. The arguments are effective. After the discussion stage, all reviews are positive regarding the paper. While the ablation study suggests that NeRF provides better performance, it remains unclear whether it is over-engineering. The paper could be stronger if it can provide better intuition.

**Justification For Why Not Higher Score:**

In spite of the fact that the ablation study indicates that NeRF provides better performance, it does not completely resolve the over-engineering issue since there is no convincing intuition.

**Justification For Why Not Lower Score:**

The proposed approach is the first unsupervised method that is capable of generalizing texture synthesis with dense 3D correspondence. By exploring the power of unsupervised learning on large 2D image collections, it is capable of generating realistic category-level textures.

---

### Decision · Program_Chairs · 2024-01-16

Accept (poster)